# Comparative Proteomic Analysis Reveals Mx1 Inhibits Senecavirus A Replication in PK-15 Cells by Interacting with the Capsid Proteins VP1, VP2 and VP3

**DOI:** 10.3390/v14050863

**Published:** 2022-04-21

**Authors:** Hang Gao, Zhaoying Xiang, Xinna Ge, Lei Zhou, Jun Han, Xin Guo, Yanhong Chen, Yongning Zhang, Hanchun Yang

**Affiliations:** Key Laboratory of Animal Epidemiology of Ministry of Agriculture and Rural Affairs, College of Veterinary Medicine, China Agricultural University, Beijing 100193, China; ghs1997@foxmail.com (H.G.); cauxzy@163.com (Z.X.); gexn@cau.edu.cn (X.G.); leosj@cau.edu.cn (L.Z.); hanx0158@cau.edu.cn (J.H.); guoxincau@cau.edu.cn (X.G.); chenyh@cau.edu.cn (Y.C.)

**Keywords:** Senecavirus A (SVA), PK-15 cells, tandem mass tags (TMT), liquid chromatography-tandem mass spectrometry (LC-MS/MS), differentially expressed proteins (DEPs), Mx1

## Abstract

As an emergent picornavirus pathogenic to pigs, Senecavirus A (SVA) can replicate in pig kidneys and proliferates well in porcine kidney epithelial PK-15 cells. Here, tandem mass tags (TMT) labeling coupled with liquid chromatography–tandem mass spectrometry (LC-MS/MS) was used to analyze the proteome dynamic changes in PK-15 cells during SVA infection. In total, 314, 697 and 426 upregulated differentially expressed proteins (DEPs) and 131, 263 and 342 downregulated DEPs were identified at 12, 24 and 36 hpi, respectively. After ensuring reliability of the proteomic data by quantitative PCR and Western blot testing of five randomly selected DEPs, Mx1, eIF4E, G6PD, TOP1 and PGAM1, all the DEPs were subjected to multiple bioinformatics analyses, including GO, COG, KEGG and STRING. The results reveal that the DEPs were mainly involved in host innate and adaptive immune responses in the early and middle stages of SVA infection, while the DEPs mainly participated in various metabolic processes in the late stage of infection. Finally, we demonstrated that Mx1 protein exerts antiviral activity against SVA by interacting with VP1 and VP2 proteins dependent on its GTPase, oligomerization and interaction activities, while Mx1 interacts with VP3 only depending on its oligomerization activity. Collectively, our study provides valuable clues for further investigation of SVA pathogenesis.

## 1. Introduction

Senecavirus A (SVA), once called Seneca Valley virus (SVV), is classified into the genus *Senecavirus* within the family *Picornaviridae* [1]. The virus has a linear, single-stranded, positive-sense RNA genome, which consists of a 5′ untranslated region (UTR), a large open reading frame (ORF), a 3′ UTR and a poly (A) tail [1]. The large ORF is first translated into a polyprotein precursor which is subsequently cleaved into a leader protein (L), four structural proteins (VP4, VP2, VP3 and VP1) and seven functional proteins (2A, 2B, 2C, 3A, 3B, 3C and 3D) [1]. Historically, SVA was serendipitously discovered during cultivating PER.C6 cells in the United States in 2002 [1]. Although SVA was detected out in Canadian pigs with vesicular diseases as early as 2007 [2], and a few SVA isolates were occasionally isolated from the U.S. pig populations almost at the same time [3], it was not until 2014 that the definite association of SVA infection with swine vesicular diseases was confirmed in the Brazilian swine herds [4]. Since then, the outbreak of SVA-caused swine vesicular diseases has also been reported in many other countries, such as China, Thailand, Colombia, and Vietnam. Clinically, SVA infection mainly causes vesicular lesions around the mouth and hooves of pigs of different ages and acute death of neonatal piglets [5,6]. The spread and prevalence of SVA in pig herds has caused considerable economic losses to the global swine industry [7].

As a newly emerging swine pathogen, the pathogenic mechanisms of SVA and the mechanisms of SVA–host interactions are still not fully clarified and warrant further exploration. Existing studies have demonstrated that SVA has a wide range of tissue tropism for multiple organs in naturally and experimentally infected piglets, which include tonsils, spleen, lungs, liver, and kidneys [8,9]. Moreover, it was also demonstrated that SVA has a strikingly broad cell tropism and is able to replicate in various cell lines, including human-derived PER.C6 and human embryonic kidney 293T (HEK293T) cells, porcine-derived swine testicle (ST) and porcine kidney epithelial (PK-15, SK-RST, SK-6 and IBRS-2) cells, as well as baby hamster kidney-21 (BHK-21) cells [10,11]. Although all these cell lines are permissive for SVA infection and replication, as an important pathogenic agent for pigs, the porcine-derived cell lines are more suitable for the research of SVA. Currently, the PK-15 cell line derived from swine renal epithelial cells has been widely using in the studies of SVA–host cell interactions involved in viral pathogenesis [10,12,13,14].

In recent years, mass spectrometry-based multiplexed proteomics in combination with various bioinformatics analyses have become a powerful tool for the large-scale and high-throughput identification, quantitation, and characterization of low-abundance proteins across a variety of biological samples [15]. Therefore, proteomics plays an important role in profiling proteome dynamic changes, host–pathogen interactions and signaling pathways during diverse pathogen infections [16,17], which will provide important clues for a better understanding of their pathogenesis. In the present study, a high-throughput quantitative proteomic approach of tandem mass tags (TMT) labeling coupled with liquid chromatography–tandem mass spectrometry (LC-MS/MS) was used to quantitatively analyze the dynamic changes of differentially expressed proteins (DEPs) in PK-15 cells in response to SVA infection. The identified DEPs were then subjected to comprehensive bioinformatics analyses, whereby we found that many interferon (IFN)-stimulated gene (ISG)-encoded proteins, in particular Mx1 and ISG15, were significantly upregulated in the early and middle stages rather than the late stage of SVA infection. Therefore, we investigated the antiviral activity of Mx1 and ISG15 proteins against SVA.

## 2. Materials and Methods

### 2.1. Cells, Virus and Antibodies

PK-15 cells were cultured with Dulbecco’s modified Eagle’s medium (DMEM; Gibco, Carlsbad, CA, USA) supplemented with 10% fetal bovine serum (FBS; Gibco), 100 U/mL penicillin, 100 μg/mL streptomycin, 5 µg/mL transferrin and 5 ng/mL selenium at 37 °C in a humidified incubator (Thermo Fisher Scientific, Waltham, MA, USA) with 5% CO_2_. HEK293T cells were cultured in a manner similar to PK-15 cells but without addition of transferrin and selenium. The SVA SDta/2018 strain (GenBank accession No. MN433300.1) was isolated from the vesicular fluids of a diseased piglet by our laboratory [18]. The monoclonal antibody (mAb) 2F5 (isotype IgG2a/κ) raised against the VP2 protein of SVA was prepared in our laboratory. Mouse anti-β-actin mAb was purchased from Proteintech Group, Inc. (Wuhan, China). Horseradish peroxidase (HRP)-conjugated goat anti-mouse or anti-rabbit IgG secondary antibodies were purchased from Zhongshan Golden Bridge Biotechnology Co., Ltd. (Beijing, China). Mouse anti-Myc and rabbit anti-HA mAbs were purchased from Sigma-Aldrich (St. Louis, MO, USA). Mouse anti-HA mAb was purchased from Medical & Biological Laboratories Co., Ltd. (Nagoya, Japan). Rabbit mAbs against Mx1, ISG15, eukaryotic initiation factor 4E (eIF4E), glucose 6 phosphate dehydrogenase (G6PD), topoisomerase I (TOP1) and phosphoglycerate mutase 1 (PGAM1) proteins were purchased from Abcam (Cambridge, UK). Alexa Fluor 488-conjugated goat anti-rabbit/mouse IgG and Alexa Fluor 594-conjugated goat anti-mouse IgG were purchased from Thermo Fisher Scientific.

### 2.2. 50% Tissue Culture Infectious Dose (TCID_50_) Assay

PK-15 cells grown to ~90% confluence in 96-well cell culture plates (Corning, NY, USA) were inoculated with 100 μL/well of 10-fold serial dilutions of SVA. Four repeat wells were inoculated with each diluent. The inocula were removed after a 1-h adsorption at 37 °C, and 200 μL/well of DMEM containing 2% FBS was added to each well. Following an additional 24 h culture, the presence of a visible cytopathic effect (CPE) in the corresponding wells was recorded, and viral titers were calculated using the Reed–Muench method [19].

### 2.3. Indirect Immunofluorescence Assay (IFA)

PK-15 cells grown to ~90% confluence in 96-well plates were mock infected or infected with SVA at a multiplicity of infection (MOI) of 5 TCID_50_ per cell. At 6, 12, 24 and 36 h post-infection (hpi), the cells were fixed with 4% paraformaldehyde for 10 min and then permeabilized with 0.1% Triton X-100 for 15 min at room temperature. After washing thrice with phosphate buffered saline (PBS; pH 7.4), the cells were blocked with 5% bovine serum albumin for 20 min at room temperature and then incubated with the SVA VP2-specific mAb 2F5 (1:1000 dilution) for 12 h at 4 °C. Upon washing with PBS, the cells were incubated with Alexa Fluor 488-conjugated goat anti-mouse IgG (1:1000 dilution) for 1 h at 37 °C. After another washing step, the cells were counterstained with 4′, 6-diamidino-2-phenylindole (DAPI) for 10 min at room temperature. The cells were observed with an Eclipse Ti-U microscope (Nikon Corp., Tokyo, Japan).

### 2.4. Protein Sample Preparation, Trypsin Digestion and TMT Labeling

PK-15 cells grown to ~90% confluence in 6-well cell culture plates were mock infected or infected with SVA at an MOI of 5 TCID_50_ per cell. At 12, 24 and 36 hpi, both mock- and SVA-infected cells were collected for protein sample preparation. Specifically, the cells in the six wells of the same plate were harvested and mixed together as a biological replicate. Three independent biological replicates were set for both mock- and SVA-infected cells at the three time points post infection. As a consequence, a total of 18 cell samples (Mock/12h/1, Mock/12h/2, Mock/12h/3, SVA/12h/1, SVA/12h/2, SVA/12h/3, Mock/24h/1, Mock/24h/2, Mock/24h/3, SVA/24h/1, SVA/24h/2, SVA/24h/3, Mock/36h/1, Mock/36h/2, Mock/36h/3, SVA/36h/1, SVA/36h/2, and SVA/36h/3) were obtained. Before cell collection, the cells were rinsed with prechilled PBS to remove the residual serum proteins of FBS, which were then harvested with disposable cell scrapers. After centrifugation (300× *g*, 10 min), the cell pellets were lysed with 800 µL of lysis buffer consisting of 8 M urea, 1% SDS and protease inhibitor cocktail (Thermo Fisher Scientific, Rockford, IL, USA). The cell lysates were then sonicated for 2 min and further incubated on ice for 30 min. After a 30-min centrifugation (12,000× *g*, 4 °C), the protein concentration of the resulting supernatants (i.e., protein samples) was determined using a Thermo Fisher Scientific Pierce^™^ BCA Protein Assay Kit (Rockford, IL, USA). For reduction and alkylation of the protein samples, a final concentration of 100 mM triethylammonium bicarbonate (TEAB) was added to 100 μg of proteins from each sample, followed by adding a final concentration of 10 mM tris (2-carboxyethyl) phosphine and maintaining reaction for 1 h at 37 °C. Afterwards, iodoacetamide was added to the protein solutions at a 40 mM final concentration. Following a 40 min incubation at room temperature under light-free conditions, precooled acetone was added to the protein solutions in a ratio of 6:1 (*v*/*v*), which were placed at −20 °C for 4 h to precipitate the proteins. After centrifugation for 20 min at 10,000× *g* and 4 °C, the precipitates from each protein sample were dissolved with 100 μL of 100 mM TEAB, and then digested with Promega modified trypsin (Madison, WI, USA) in a ratio of 50:1 (*m*/*m*) overnight at 37 °C to generate peptides. The tryptic peptides were then labeled using a TMT 10plex Isobaric Label Reagent Set (Thermo Fisher Scientific, Rockford, IL, USA) according to the manufacturer’s protocol. Briefly, the tryptic peptides of the 18 protein samples were divided to two groups. Group 1 included the peptides of SVA/12h/1, Mock/12h/1, SVA/24h/1, Mock/24h/1, Mock/36h/1, SVA/36h/1, SVA/12h/3, SVA/24h/3 and SVA/36h/3, which were labeled with TMT10-126, TMT10-127N, TMT10-127C, TMT10-128N, TMT10-128C, TMT10-129N, TMT10-129C, TMT10-130N, and TMT10-130C, respectively. Group 2 included the peptides of SVA/12h/2, Mock/12h/2, SVA/24h/2, Mock/24h/2, Mock/36h/2, SVA/36h/2, Mock/12h/3, Mock/24h/3 and Mock/36h/3, which were labeled with TMT10-126, TMT10-127N, TMT10-127C, TMT10-128N, TMT10-128C, TMT10-129N, TMT10-129C, TMT10-130N, and TMT10-130C, respectively. Equal amounts of labeled peptides of each sample in the same group were combined into a new microcentrifuge tube, and dried by vacuum centrifugation.

### 2.5. High-pH Reversed-Phase Liquid Chromatography Fractionation

The labeled peptides were fractionated by a Thermo Fisher Scientific Vanquish Duo UHPLC system coupled with an ACQUITY UPLC BEH C18 Column (300 Å, 1.7 μm, 2.1 mm × 150 mm; Waters Corp., Milford, MA, USA). Briefly, the mixed peptides from each group were reconstituted with mobile phase A (2% acetonitrile; adjusted to pH 10 with 0.05% ammonia), and then loaded onto the column. Peptide fractionation was performed using a linear gradient of mobile phase B (80% acetonitrile; adjusted to pH 10 with 0.05% ammonia): 0−1.9 min, 0% B; 1.9−2 min, 0−5% B; 2−17 min, 5% B; 17−18 min, 5−10% B; 18−35.5 min, 10−30% B; 35.5−38 min, 30−36% B; 38−39 min, 36−42% B; 39−40 min, 42−100% B; 40−44 min, 100% B; 44−45 min, 100−0% B; 45−48 min, 0% B. The column flow was maintained at a flow rate of 200 μL/min and monitored by measuring absorbance at 214 nm. Finally, a total of 10 fractions were obtained by pooling two of the 20 fractions and vacuum dried for subsequent LC-MS/MS analysis.

### 2.6. LC-MS/MS Analysis

Two micrograms of the labeled peptides from each fraction were dissolved with mobile phase A (2% acetonitrile, 0.1% formic acid), and then loaded onto a C18 Reversed Phase HPLC Column (75 μm × 25 cm; Thermo Fisher Scientific). Chromatographic separation of the peptides was performed on an EASY-nLC 1200 Ultra-performance liquid chromatography (UPLC) system (Thermo Fisher Scientific) at a flow rate of 300 nL/min over a gradient of 5−23% mobile phase B (80% acetonitrile, 0.1% formic acid) for 64 min, 23−29% B for 16 min, 29−38% B for 10 min, 38−48% B for 2 min, 48−100% B for 1 min, and then holding at 100% B for the last 27 min. The separated peptides were then subjected to nanoelectrospray ionization source followed by tandem mass spectrometry in a Q Exactive HF-X Mass Spectrometer (Thermo Fisher Scientific) coupled online to the HPLC. MS was operated in the data-dependent acquisition mode by which the MS1 scans were acquired at a resolution of 60,000, an automatic gain control (AGC) of 3E6, a maximum injection time of 20 ms, and a scan range of 350–1300 mass/charge ratio (*m*/*z*). The top 20 most intense parent ions were selected for secondary fragmentation using the higher-energy collisional dissociation method. The MS2 scans were acquired using the following parameters: resolution 45,000; AGC 2E5; maximum injection time 96 ms; fixed first mass 100 *m*/*z*; minimum AGC target 8E3; intensity threshold 8.3E4; and dynamic exclusion time 20 s.

### 2.7. Data Availability

The obtained mass spectrometry proteomics data have been deposited to the ProteomeXchange Consortium (http://www.proteomexchange.org, accessed on 26 January 2022) via the Proteomics Identifications (PRIDE) [20] partner repository with the dataset identifier PXD031260.

### 2.8. Database Search

The obtained MS/MS raw data were searched against the UniProt *Sus scrofa* and Senecavirus A database for protein identification and quantification using the Proteome Discoverer Software version 2.4 (Thermo Fisher Scientific, San Jose, CA, USA). The main search parameters were as follows: cysteine alkylation, iodoacetamide; dynamic modification, oxidation (M), acetyl (protein N-terminus), Met-loss (protein N-terminus), Met-loss+Acetyl (protein N-terminus); static modification, carbamidomethyl (C), TMT 6plex (K), TMT 6plex (N-terminus); enzyme name, trypsin (full); maximal missed cleavage sites, 2; precursor mass tolerance, 20 ppm; fragment mass tolerance, 0.02 Da; validation based on, *Q*-value. The false discovery rate for peptide and protein identifications was set as ≤1%. Protein quantification data with a fold change (FC) > 1.20 or < 0.83 and a *p* value < 0.05 between two comparable samples were set as the significance threshold for DEPs.

### 2.9. Bioinformatics Analysis

The identified DEPs were subjected to a series of bioinformatics analyses as previously described [21]. First, gene ontology (GO) analysis was performed to divide the DEPs into three categories of biological process (BP), cellular component (CC) and molecular function (MF). To further investigate which DEPs were significantly enriched in the GO terms, GO enrichment analysis was conducted using the online Goatools (https://github.com/tanghaibao/Goatools, accessed on 27 October 2020). Second, for a more comprehensive functional annotation, the DEPs were searched against the Clusters of Orthologous Groups of proteins (COG) database (http://www.ncbi.nlm.nih.gov/COG/, accessed on 27 October 2020) [22]. Third, to predict the potential signal pathways related to the DEPs, the Kyoto Encyclopedia of Genes and Genomes (KEGG) pathway analysis was carried out using DIAMOND BLASTP against the KEGG database (http://www.genome.jp/kegg/, accessed on 27 October 2020) with a cutoff E-value ≤ 1 × 10^−5^. Moreover, KEGG pathway enrichment analysis was also performed on the DEPs using the KOBAS software (http://kobas.cbi.pku.edu.cn/home.do/, accessed on 27 October 2020). For both GO and KEGG enrichment analyses, a Fisher’s exact test was used to identify the enriched terms, with *p* values < 0.05 considered statistically significant. Finally, the protein–protein interaction (PPI) networks between the screened DEPs were created using the Search Tool for the Retrieval of Interacting Genes/Proteins (STRING) database (http://string-db.org/, accessed on 27 October 2020) [23].

### 2.10. Quantitative Real-Time PCR (qPCR)

Total cellular RNAs of both mock- and SVA-infected PK-15 cells were extracted using MagZol Reagent (Guangzhou Magen Biotechnology Co., Ltd., Guangzhou, China) according to the manufacturer’s instructions. Two micrograms of cellular RNA from each sample were used to prepare the first-strand cDNA using a FastQuant RT Kit (Tiangen Biotech Co. Ltd., Beijing, China) as per the manufacturer’s protocol. The prepared cDNA was then used as a template in the subsequent qPCR assays to evaluate the mRNA expression levels of five randomly selected representative DEPs, including two upregulated DEPs, Mx1 and TOP1, and three downregulated DEPs, eIF4E, G6PD and PGAM1. The qPCR assays were operated on an ABI 7500 Real-Time PCR System (Applied Biosystems, Foster City, CA, USA) following the protocol of the SYBR Select Master Mix (Thermo Fisher Scientific). Porcine β-actin was used as a housekeeping gene to normalize the target gene transcript levels [21]. The relative mRNA expression levels of the selected DEPs were calculated using the 2^−ΔΔCt^ method [24]. The primers used for the qPCR assays are listed in Table 1.

### 2.11. Plasmid Construction

The coding sequences of porcine Mx1, Mx2, ISG15, OASL and IFIT1 genes were amplified by PCR using the prepared cDNA from PK-15 cells as the template and with the corresponding primer pairs listed in Table 1. The amplicons were cloned into the EcoR I/Xho I-linearized pCMV-Myc-N vector (Clontech Laboratories Inc., Palo Alto, CA, USA) by means of a homologous recombination technique using a ClonExpress II One Step Cloning Kit (Vazyme Biotech Co., Ltd., Nanjing, China) according to the manufacturer’s instructions. The resulting plasmids were designated pCMV-Myc-Mx1, pCMV-Myc-Mx2, pCMV-Myc-ISG15, pCMV-Myc-OASL and pCMV-Myc-IFIT1, respectively. Furthermore, three mutant plasmids, pCMV-Myc-Mx1(K83A), pCMV-Myc-Mx1(R409D) and pCMV-Myc-Mx1(ΔL4), were constructed using pCMV-Myc-Mx1 as the backbone plasmid and with the respective mutagenic primer pairs listed in Table 1. The three Mx1 mutants (K83A, R409D and ΔL4) contained a single-site mutation at positions 83 (K→A) and 409 (R→D), and a 40-amino acid deletion at positions 534–573, respectively. In addition, the coding sequences of L-VP4, VP1, VP2, VP3, 2A-2B, 2C, 3A-3B, 3C and 3D genes were amplified by RT-PCR using the SVA SDta/2018 RNA as the template and with the primer pairs listed in Table 1. The amplicons were cloned into the EcoR I/Xho I-linearized pCMV-HA vector (Clontech) in the same way as pCMV-Myc-N. The resulting plasmids were designated pCMV-HA-L-VP4, pCMV-HA-VP1, pCMV-HA-VP2, pCMV-HA-VP3, pCMV-HA-2A-2B, pCMV-HA-2C, pCMV-HA-3A-3B, pCMV-HA-3C, and pCMV-HA-3D, respectively. The PCR and RT-PCR amplifications were performed using the Invitrogen Platinum SuperFi II Green PCR Master Mix and SuperScript IV One-Step RT-PCR System, respectively, following the manufacturer’s protocol. All the constructed plasmids were confirmed by DNA sequencing to ensure their accuracy.

### 2.12. Western Blot (WB)

Cells grown in 6-well cell plates were harvested at the specific time points post-infection or post-transfection indicated in the figures or figure legends. After washing twice with PBS, the cells were lysed with 200 μL/well of NP-40 lysis buffer (Beyotime, Shanghai, China) containing a protease inhibitor cocktail (Thermo Fisher Scientific) for 30 min on ice. The cell lysates were centrifugated for 30 min at 12,000× *g* and 4 °C, and protein concentration in the supernatant was determined using a Pierce BCA Protein Assay Kit (Thermo Fisher Scientific). Approximately, 20 µg/lane of each protein sample was separated on 12% sodium dodecyl sulfate–polyacrylamide gel electrophoresis (SDS-PAGE) gels under reducing conditions. The separated proteins in the gel were then electrically transferred onto 0.22 μm polyvinylidene fluoride membranes (Millipore, Bedford, MA, USA). After blocking with 5% nonfat dry milk for 2 h at room temperature, the membranes were probed with appropriate primary antibodies for 1 h at 37 °C. After thorough washing with PBST (PBS containing 0.05% Tween-20, pH 7.4), the membranes were incubated with the corresponding HRP-conjugated secondary antibodies for 1 h at 37 °C. Following another wash step, the immunoreactive protein bands on the membranes were developed with the Enlight^™^ Western Blotting Substrate (Engreen Biosystem Co. Ltd., Beijing, China), and images were taken using the ChemiDoc^™^ MP Imaging System (Bio-Rad Laboratories Inc., Hercules, CA, USA).

### 2.13. RNA Interference

Three pairs of small interfering RNAs (siRNAs) targeting different regions of porcine Mx1 gene were designed to specifically knock down Mx1 expression in PK-15 cells (Table 2). Moreover, a set of siRNA duplexes previously designed for the specific knockdown of ISG15 gene expression were synthesized as well [25]. These specific siRNAs along with the scrambled siRNAs (siNC) were synthesized by Suzhou GenePharma Co., Ltd. (Suzhou, China). PK-15 cells grown to ∼50% confluence in 6-well cell plates were transfected with a pool of three siRNAs specific for Mx1 gene (40 pmol for each siRNA), 80 pmol of siISG15 or 80 pmol of siNC using Lipofectamine^®^ 3000 reagent (Invitrogen; Carlsbad, CA, USA) as per the manufacturer’s protocol. At 36 h post-transfection (hpt), the transfected cells were mock infected or infected with SVA at an MOI of 0.1 or 1. At 24 hpi, the cells and progeny viruses were harvested for WB analysis and viral yield titration, respectively.

### 2.14. Co-Immunoprecipitation (Co-IP) and Confocal Immunofluorescence Microscopy

HEK293T cells grown to ∼50% confluence in 6-well cell plates were co-transfected with 1.25 μg of pCMV-HA-L-VP4, pCMV-HA-VP1, pCMV-HA-VP2, pCMV-HA-VP3, pCMV-HA-2A-2B, pCMV-HA-2C, pCMV-HA-3A-3B, pCMV-HA-3C, pCMV-HA-3D or pCMV-HA, and 1.25 μg of pCMV-Myc-Mx1, pCMV-Myc-Mx1(K83A), pCMV-Myc-Mx1(R409D), pCMV-Myc-Mx1(ΔL4), pCMV-Myc-ISG15 or pCMV-Myc-N using Lipofectamine^®^ LTX reagent (Invitrogen) according to the manufacturer’s instruction. At 36 hpt, the cells were processed for Co-IP and confocal microscopy, respectively. For Co-IP analysis, the cells were lysed with 200 μL/well of NP-40 lysis buffer supplemented with a protease inhibitor cocktail (Thermo Fisher Scientific). After centrifuging the cell lysates for 10 min at 10,000× *g* and 4 °C, the supernatants were collected and subjected to Co-IP analyses. Briefly, 300 μL of supernatant were incubated with 2 μg of anti-Myc mAb (Sigma-Aldrich) or anti-HA mAb (Medical & Biological Laboratories Co., Ltd.) for 12 h at 4 °C with shaking. After pre-washing with Tris-buffered saline containing 0.05% Tween-20 (TBST) with a magnetic stand, 25 µL of Pierce^™^ Protein A/G Magnetic Beads (Thermo Fisher Scientific) were added to each of the antigen–antibody mixtures, and then mixed by rocking for 1 h at room temperature. After three successive washes with TBST and a final wash with ultrapure water, the immune complexes were separated on 12% SDS-PAGE gels and then analyzed by WB analysis. For confocal microscopy analysis, the cells were fixed, permeabilized and then subjected to immunofluorescent double staining using mouse anti-Myc and rabbit anti-HA mAbs (Sigma-Aldrich) as the primary antibodies, along with Alexa Fluor 488-conjugated goat anti-rabbit IgG and Alexa Fluor 594-conjugated goat anti-mouse IgG as the secondary antibodies. After counterstaining with DAPI, cell images were taken using a Nikon A1 confocal laser scanning microscope.

### 2.15. GTPase Activity Assay

The GTPase activity of wild-type (WT) Mx1 protein and its mutants Mx1(K83A), Mx1(R409D) and Mx1(ΔL4) expressed in HEK293T cells was measured as previously described [25], with slight modifications. Briefly, HEK293T cells grown to ∼50% confluence in 6-well plates were transfected with 2.5 μg/well of pCMV-Myc-Mx1, pCMV-Myc-Mx1(K83A), pCMV-Myc-Mx1(R409D), pCMV-Myc-Mx1(ΔL4), pCMV-Myc-ISG15, or pCMV-Myc-N using Lipofectamine^®^ LTX reagent. At 36 hpt, the cells were rinsed once with ddH_2_O, scraped off, and pelleted by centrifugation at 300× *g* for 6 min. After three washes with ddH_2_O, the cells were suspended in 500 μL ddH_2_O and lysed by sonication on ice for 2 min at 1 s pulses and 9 s intervals. The cell lysates were centrifuged for 30 min at 10,000× *g* and 4 °C, and the supernatants were subjected to GTPase activity assay with a QuantiChrom^™^ ATPase/GTPase Assay Kit (BioAssay Systems, Hayward, CA, USA) following the manufacturer’s protocol. The optical density at 620 nm (OD_620_) was measured using a Tecan Spark Multimode Microplate Reader (Männedorf, Switzerland). The relative enzyme activities of the Mx1 mutants were calculated with the OD_620_ values and were expressed as relative to the activity of the Mx1 (WT) protein that was defined as 100%.

### 2.16. Statistical Analysis

Experimental data were presented as means ± standard deviation (SD) and were analyzed using GraphPad Prism software (Version 8.0; La Jolla, CA, USA). Except GO and KEGG enrichment which were analyzed by Fisher’s exact test, the other data were analyzed by Student’s *t* test. Differences were considered statistically significant at *p* values of <0.05 (*), <0.01 (**), <0.001 (***), and <0.0001 (****).

## 3. Results

### 3.1. Determination of the Optimal Time Points for Proteomic Analysis Following SVA Infection

To screen the optimal time points for the proteomic analysis, PK-15 cells were mock-infected or infected with the SVA SDta/2018 strain at an MOI of 5 TCID_50_ per cell, and then microscopically monitored for the presence of CPE. Meanwhile, the proliferation kinetics of SVA in PK-15 cells was assessed by determining the viral titers at 6, 12, 24, 36 and 48 hpi. Compared with the mock-infected cells, although no CPE was observed in SVA-infected cells at 6 hpi, visible CPEs gradually became apparent as infection progressed (Figure 1A), which were characterized by formation of cell clusters, detachment and disintegration of cells. The propagation of SVA in PK-15 cells at the specified time points was further confirmed by IFA using an mAb (2F5) that specifically recognizes the SVA VP2 protein (Figure 1B). The one-step growth curve of SVA further demonstrated that the virus titer peaked at 36 hpi after which it gradually declined (Figure 1C). From 48 hpi onwards, the majority of SVA-infected cells disintegrated and became detached from the monolayer (Figure 1A). This will inevitably lead to the release of cellular proteins into the culture medium, thus causing the proteome to be underestimated. For the above reasons, we decided to choose 12, 24 and 36 hpi as the optimal sampling time for further proteomic analysis with the purpose of clarifying the dynamic changes of DEPs as SVA infection progressed.

### 3.2. Temporal Proteomic Analysis of PK-15 Cells in Response to SVA Infection

To investigate the dynamic changes in the proteome of PK-15 cells during the course of SVA infection, the total proteins prepared from mock- and SVA-infected PK-15 cells at an early, medium and late time point of infection were analyzed by TMT-labeling coupled with LC-MS/MS. Overall, 8512 proteins, including the polyprotein, VP0 and VP1 proteins of SVA, were identified in both mock- and SVA-infected PK-15 cells at 12, 24 and 36 hpi (Appendix A). After filtering with the widely used cutoff criteria of FC > 1.2 or <0.83 and *p* values < 0.05 [21,26], among the 8512 proteins, 314, 697 and 426 upregulated proteins and 131, 263 and 342 downregulated proteins were identified as DEPs at 12, 24 and 36 hpi, respectively. Obviously, the number of DEPs exhibited an increasing trend as SVA infection progressed. However, due to the inadequate functional annotation of pig genomes, 85, 225 and 158 uncharacterized DEPs were screened at 12, 24 and 36 hpi, respectively (Appendix A).

### 3.3. Validation of the Proteomics Data by qPCR and WB

To validate the obtained proteomics data, qPCR was performed to determine the transcription levels of five randomly selected representative DEPs (Mx1, TOP1, eIF4E, G6PD and PGAM1) in PK-15 cells in response to SVA infection. As shown in Figure 2A, the relative mRNA expression levels of Mx1 and TOP1 were significantly upregulated, whereas those of eIF4E, G6PD and PGAM1 were significantly downregulated in PK-15 cells during infection with SVA. For further validation, WB was conducted to detect the protein expression levels of the five DEPs, and the results indicated that their change tendencies were similar to their respective mRNA levels (Figure 2B). Further quantitative analyses of the relative mRNA and protein expression ratios of the selected five DEPs showed that their changing trends are consistent with the MS data (Figure 2C). Taken together, our quantitative proteomics data obtained by the TMT-coupled LC-MS/MS strategy are reliable and effective, and thus suitable for the subsequent analyses.

### 3.4. GO Functional Annotation of the DEPs

To characterize the DEPs identified at 12, 24 and 36 hpi, all the DEPs were searched against the GO database to annotate them under the three major categories of BP, CC and MF. The top 20 GO terms with the most DEPs were selected from the three categories and are shown in Figure 3A. Within the BP category, the relevant DEPs mainly participated in 13 biological processes, including cellular process, biological regulation, metabolic process, response to stimulus, localization, immune system process, developmental process, interspecies interaction between organisms, biological adhesion, etc. Within the CC category, the relevant DEPs were annotated to be distributed within only two cellular components, cellular anatomical entity and protein-containing complex; within the MF category, the relevant DEPs were mainly involved in seven molecular functions, including binding, catalytic activity, molecular function regulator, transporter activity, transcription regulator activity, structural molecule activity, and molecular transducer activity (Figure 3A; Appendix A). Notably, the number of upregulated DEPs at 24 hpi was the highest among the three time points post-infection (12, 24 and 36 hpi) in most GO terms, whereas the highest number of downregulated DEPs appeared at 36 hpi. To further characterize the DEPs in depth, GO term enrichment analysis was performed and the top 20 significantly enriched GO terms within the three categories were selected from 12, 24 and 36 hpi. As shown in Figure 3B, the relevant DEPs were significantly enriched in 33, 8 and 3 GO terms of BP, CC and MF, respectively (Appendix A). To be specific, the top 20 significantly enriched GO terms at 12 hpi were involved in 20 out of the 33 BP subclasses, such as negative regulation of viral genome replication, defense response to other organisms, immune response, immune effector process, defense response to virus, negative regulation of viral life cycle, response to type I interferon, and antigen processing and presentation of endogenous antigens, but they did not involve the CC and MF categories. Interestingly, the top 20 significantly enriched GO terms at 24 hpi shared 16 identical BP subclasses with those at 12 hpi and occupied a unique BP subclass (regulation of response to stress) and three CC subclasses (intrinsic component of membrane, integral component of membrane and mitochondrial membrane). In contrast, the top 20 significantly enriched GO terms at 36 hpi covered all the three major GO categories, but the involved subclasses of BP, CC and MF were completely different from those at 12 hpi and 24 hpi. The DEPs corresponding to the significantly enriched GO terms at 36 hpi were mainly involved in a variety of metabolic processes in the BP category, such as the carbohydrate metabolic process, carboxylic acid metabolic process, small molecule metabolic process, oxoacid metabolic process, hormone metabolic process, organic acid metabolic process, monocarboxylic acid metabolic process, regulation of the ATP metabolic process, and positive regulation of the steroid metabolic process. Moreover, the relevant DEPs at 36 hpi were significantly enriched in five GO terms of CC and three GO terms of MF. Taken together, these data reveal that the significantly enriched DEPs were mainly related to host immune responses; in particular, the innate immune response, in the early and middle stages of SVA infection, but they were mainly involved in various metabolic processes in the late stage.

### 3.5. COG Functional Classification of the DEPs

To annotate the protein function, the DEPs were searched against the COG database. As shown in Figure 4, all the DEPs identified at 12, 24 and 36 hpi were classified into 21 categories based on the COG functional annotation. The top five categories containing more than 25 DEPs were intracellular trafficking, secretion and vesicular transport; posttranslational modification, protein turnover and chaperones; translation, ribosomal structure and biogenesis; transcription; and function unknown (Appendix A). Furthermore, many of the DEPs participated in multiple biological and metabolic processes, including amino acid transport and metabolism, nucleotide transport and metabolism, carbohydrate transport and metabolism, coenzyme transport and metabolism, lipid transport and metabolism, inorganic ion transport and metabolism, energy production and conversion, and secondary metabolite biosynthesis, transport and catabolism. The remaining DEPs were mainly involved in the COG categories of signal transduction mechanisms, defense mechanisms, cytoskeleton, etc. Similar to the GO annotation, the number of DEPs at 24 hpi was the highest among the three time points post-infection in most COG terms. Disappointingly, 193, 419 and 314 DEPs identified at 12, 24 and 36 hpi, respectively, were classified into the same COG category with function unknown.

### 3.6. KEGG Pathway Analysis of the DEPs

To investigate the underlying signaling pathways in which the DEPs were involved, KEGG pathway analyses were performed. As shown in Figure 5A–C, the relevant DEPs identified at 12, 24 and 36 hpi were all classified into six first-category KEGG pathways, including metabolism, genetic information processing, environmental information processing, cellular processes, organismal systems, and human diseases (Appendix A), each of which contained diverse second-category KEGG pathways. Within the metabolism category, the top 5 pathways with the most DEPs at 12, 24 and 36 hpi were mainly concentrated in global and overview maps, carbohydrate metabolism, amino acid metabolism, lipid metabolism, and metabolism of cofactors and vitamins. Within the genetic information processing category, the relevant DEPs at 12, 24 and 36 hpi were mainly involved in 4 signaling pathways, including folding, sorting and degradation, translation, replication and repair, and transcription. Within the environmental information processing category, the relevant DEPs at 12, 24 and 36 hpi were primarily involved in 3 signaling pathways, including signal transduction, signaling molecules and interaction, and membrane transport. Within the cellular processes category, the relevant DEPs at 12, 24 and 36 hpi were mainly related to cell growth and death, transport and catabolism, cell motility, and cellular community—eukaryotes. Within the organismal systems category, the top 5 pathways with the most DEPs at 12, 24 and 36 hpi were mainly involved in the immune system, endocrine system, environmental adaptation, digestive system, and nervous system; and within the human diseases category, the top 5 pathways with the most DEPs at 12, 24 and 36 hpi were mainly involved in viral infectious disease, cancer overview, bacterial infectious disease, endocrine and metabolic disease, and neurodegenerative disease. Taken together, the number of DEPs within most of the KEGG categories increased in the early stage of SVA infection but decreased in the late stage, with the peak appearing at *24* hpi. To explore the signaling pathways in depth, the DEPs were additionally subjected to KEGG pathway enrichment analysis. The top 50 DEPs in the number of annotated KEGG signaling pathways whose enrichment *p* values ranked within the top 15 were extracted from the three time points and shown in Figure 5D–F. The relevant DEPs identified at 12 hpi were mainly enriched in virus infection-related pathways (such as influenza A, herpes simplex virus 1 infection, hepatitis C, measles, Epstein–Barr virus infection, and human papillomavirus infection), innate immune response-related pathways (such as the NOD-like receptor signaling pathway, RIG-I-like receptor signaling pathway, complement and coagulation cascades, and TNF signaling pathway), and other important signaling pathways, including primary immunodeficiency, intestinal immune network for IgA production, cytokine–cytokine receptor interaction, malaria, and ECM-receptor interaction (Figure 5D and Appendix A). For the DEPs identified at 24 hpi, most of them were still enriched in virus infection-related pathways and innate immune response-related pathways, with a new member of the cytosolic DNA-sensing pathway identified in the latter. Moreover, many DEPs were enriched in several new signaling pathways, such as cell adhesion molecules, neuroactive ligand–receptor interaction, autoimmune thyroid disease, allograft rejection, and non-alcoholic fatty liver disease (Figure 5E and Appendix A). By contrast, the majority of the DEPs identified at 36 hpi were mainly enriched in multiple metabolism-related signaling pathways, such as glycolysis/gluconeogenesis, cysteine and methionine metabolism, chemical carcinogenesis, drug metabolism—cytochrome P450, metabolism of xenobiotics by cytochrome P450, drug metabolism—other enzymes, the PPAR signaling pathway, biosynthesis of amino acids, type I diabetes mellitus, carbon metabolism, peroxisome, and glycine, serine and threonine metabolism. Interestingly, except complement C3 (UniProt accession No. I3LTB8) enriched in the complement and coagulation cascades pathway, no other DEPs at 36 hpi were enriched in virus infection-related pathways and innate immune response-related pathways (Figure 5F and Appendix A). Furthermore, except the non-alcoholic fatty liver disease pathway annotated at 24 hpi, the Z-scores of all the other KEGG signaling pathways at 12 and 24 hpi were greater than zero. This indicates that the number of upregulated DEPs involved in these signaling pathways was more than that of downregulated DEPs, suggesting that the relevant pathways were activated. On the contrary, except the complement and coagulation cascades pathway, the Z-scores of KEGG signaling pathways at 36 hpi were less than zero, revealing that the relevant pathways were inhibited (Figure 5D–F).

### 3.7. PPI Network Analysis of the DEPs

To further explore the potential functions and interactions of the dysregulated DEPs, PPI network analyses were performed. For the DEPs at 12 hpi, they were mainly mapped to three major strongly interacting functional networks, which were composed of three groups of strongly interacting proteins, including the ISG family proteins, ISG15-OASL-ZNFX1-GBP1-DHX58-IFI44-IFI44L-IFIT3-IRF9-UBE2N-OAS1-OAS2, the endogenous antigen processing and presentation signaling pathway related proteins, PSMB9-DTX3L-PSMA6-TAPBPL-ENSSSCP00000001587(TAP2)-TAPBP-PSMG1, and the multifunctional proteins regulating cell growth and differentiation, TGFB1-FGF2-CCN2-SPARC. There exist at least three proteins acting as hub proteins in these three tightly connected networks, including ISG15, PSMB9 and TGFB1 (Figure 6A and Appendix A). For the DEPs at 24 hpi, there are at least five groups of proteins with strong interaction, including ISG15-OASL-DHX58-IFI35-IFI44L-OAS1-OAS2-IRF9-IFI44-GBP1-IFIT3-ZNFX1, which are involved in innate immunity; RPP38-TACSTD2-RPL21-RPL19-ENSSSCP00000020253(RPS17)-ENSSSCP00000008721(RPL31)-RTCB-SF3B2, which are associated with mRNA translation, mRNA splicing, protein synthesis, and posttranslational protein folding; HSPA5-DNAJA1-DPY30-UBQLN1-ENSSSCP00000020707(CCT6)-SETD7, which are related to protein folding, assembly and modification; COX2-COX6A1-SDHC-COX18-UQCR10-NDUFA4-COX17-UQCR10-NDUFB1-NDUFAF3, which are mainly involved in the mitochondrial respiratory chain; and CCNB1-TGFB1-MAP2K7-XIAP-KIF11-AURKA-TACC3-GTSE1-ANAPC4-PARPBP-RIPK1, which are associated with cell cycle and cellular components (Figure 6B and Appendix A). For the DEPs at 36 hpi, they were mainly mapped to four relatively independent functional networks, including KRR1-NOB1-DHX37-MRTO4-IMP3-NSA2-TRMT112-RRP1B-ABT1-NOL10-CEBPZ, which are involved in ribosome biogenesis, ribosomal RNA processing and ribosome assembly; CCNB1-KPNA2-KIF23-TOPOII-TOP1MT-STMN1-MELK-MKI67, which are involved in cell cycle regulation and nuclear protein import; GSTT1-GSTP1-MGST3-PRDX6-PGK1-PGAM1-GPI-HPRT1-GSTK1-ALDH3B1-INS, which mainly participate in energy metabolism and lipid metabolism; and ACOX1-ACSL4-SCP2-ACAA2-DECR1-ECH1-LIPG, which are mainly associated with fatty acid metabolism (Figure 6C and Appendix A). Taken together, as the infection progressed, the proteome of SVA-infected PK-15 cells changed dynamically, with most of the significantly enriched DEPs mainly related to host immune responses in the early and middle stages of SVA infection, but mainly involved in various metabolic processes in the late stage. Notably, not all DEPs exhibited connection with others because their functions were either unrelated or have not yet been annotated.

### 3.8. The ISG Family Proteins Inhibit SVA Replication

Since the above proteomic data and bioinformatics analyses demonstrated that a variety of ISG family proteins (such as Mx1, Mx2, IFIT1, ISG15 and OASL) were significantly upregulated in the early and middle stages of SVA infection, we next focused on assessing their impact on SVA replication. To this end, five representative recombinant pCMV-Myc-N plasmids, respectively, expressing Myc-tagged Mx1, Mx2, IFIT1, ISG15 and OASL were separately transfected into PK-15 cells. At 36 hpt, the transfected cells were mock-infected or infected with SVA at a low MOI of 0.1 and a high MOI of 1. At 24 hpi, the cells were harvested and processed for WB analysis. As shown in Figure 7A, compared to the empty vector pCMV-Myc-N-transfected cells and non-transfected cells, the expression level of VP2 protein was not detectable in SVA-infected PK-15 cells which had been pre-transfected with pCMV-Myc-Mx1, pCMV-Myc-Mx2, pCMV-Myc-IFIT1, pCMV-Myc-ISG15 or pCMV-Myc-OASL. This suggests that overexpression of the ISG proteins in PK-15 cells decreased SVA replication, which was not affected by the inoculation dose of SVA. To backward verify the antiviral function of the relevant ISG proteins, the endogenous Mx1 and ISG15 proteins in PK-15 cells were specifically knocked down by RNA interference, and then their abilities to support SVA replication were evaluated by WB analysis of VP2 expression and TCID_50_ assay of progeny virus yield. As shown in Figure 7B,C, knockdown of ISG15 and Mx1 proteins in PK-15 cells enhanced the expression of SVA VP2 protein and the viral titers of SVA. Collectively, these results reveal that Mx1 and ISG15 are able to effectively inhibit the replication of SVA.

### 3.9. Mx1 but Not ISG15 Interacts with SVA VP1, VP2 and VP3 Proteins

A number of previous studies have demonstrated that Mx1 and ISG15 proteins exert their antiviral activity by interacting with specific proteins of the virus [25,27,28]. To find out whether there exist interactions between Mx1, ISG15 and SVA proteins, Co-IP assays were performed on the cell lysates prepared from 293T cells co-transfected with recombinant pCMV-HA plasmids expressing SVA proteins and pCMV-Myc-Mx1 or pCMV-Myc-ISG15. As shown in Figure 8A, only the HA-tagged VP1, VP2 and VP3 proteins rather than other SVA proteins were co-immunoprecipitated with the Myc-tagged Mx1 protein, suggesting that Mx1 interacts with the SVA VP1, VP2 and VP3 proteins. To confirm the interactions, Co-IP was conducted once again using the HA mAb to pull down the immune complexes, which were tested to be simultaneously positive for both HA and Myc mAbs (Figure 8B), indicating that Mx1 indeed interacts with the VP1, VP2 and VP3 proteins of SVA. Notably, although ISG15 is also able to inhibit SVA replication, the Co-IP results showed that it does not interact with the VP1, VP2 and VP3 proteins (Figure 8C). As an additional confirmatory assay, confocal microscopy was also performed on the co-transfected 293T cells. As shown in Figure 9, the green immunostaining signals corresponding to the HA-tagged VP1, VP2 and VP3 proteins were well co-localized with the red immunostaining signals corresponding to the Myc-tagged Mx1 protein in the cytoplasm of co-transfected 293T cells. By contrast, no co-localizing signals were observed between the immunostaining signals of other SVA proteins (such as L-VP4, 2A-2B, 2C, 3A-3B, 3C and 3D) and those of the Myc-tagged Mx1 protein. Taken together, these results demonstrate that Mx1 protein exerts antiviral activity against SVA by interacting with the VP1, VP2 and VP3 proteins. Although ISG15 also functions to inhibit SVA replication, it does not interact with any SVA proteins.

### 3.10. Mx1-Associated GTPase, Oligomerization and Interaction Activities Are Required for Its Interaction with VP1, VP2 and VP3 Proteins

Existing studies have proven that Mx1 protein possesses three important activities, including GTPase activity, oligomerization activity and interaction activity, each of which plays a vital but not identical role in combating with different viruses [25,29,30]. For example, all the three GTPase, oligomerization and interaction activities of Mx1 are necessary for its interaction with NS5B and for its antiviral activity against classical swine fever virus (CSFV), whereas the GTPase activity of Mx1 is not essential for Mx1 to inhibit HBV replication [29]. To clarify whether all the three activities of Mx1 affect its antiviral activity against SVA, we constructed GTPase-deficient Mx1 (K83A), oligomerization-disrupting Mx1 (R409D), and interaction-interfering Mx1 (ΔL4) (Figure 10A), which have been demonstrated to completely lose anti-CSFV activity [25], and then evaluated their ability to interact with the SVA VP1, VP2 and VP3 proteins. As shown in Figure 10B, compared with the Myc-tagged WT Mx1 protein produced by transfection with pCMV-Myc-Mx1, the three mutants Mx1 (K83A), Mx1 (R409D) and Mx1 (ΔL4) completely lost their ability to interact with the HA-tagged VP1 protein produced by transfection with pCMV-HA-VP1. Likewise, the three Mx1 mutants also lost their ability to interact with the HA-tagged VP2 protein (Figure 10C). In contrast, the mutants Mx1 (K83A) and Mx1 (ΔL4) retained their ability to interact with the HA-tagged VP3 protein, whereas Mx1 (ΔL4) did not (Figure 10D). Notably, to ensure that the K83A mutation did affect the GTPase activity of Mx1 protein, a commercial kit was used to detect the GTPase activity of HEK293T cells transfected with Mx1 (WT) and mutant Mx1 (K83A), Mx1 (R409D) and Mx1 (ΔL4). As shown in Figure 10E, the GTPase activity of Mx1 (K83A) was significantly lower than those of Mx1 (WT), Mx1 (R409D) and Mx1 (ΔL4), indicating that the GTPase activity of Mx1 was successfully abolished by the K83A mutation. Taken together, these results indicate that the GTPase, oligomerization and interaction activities of Mx1 are crucial for interacting with VP1 and VP2 proteins, while only the oligomerization activity of Mx1 is required for interacting with VP3 protein.

## 4. Discussion

Although SVA was discovered more than two decades ago [1], it was not until recent years that the harm of SVA infection to the pig industry has gradually appeared across the world [5,6,31]. What caused a delayed identification of SVA as a causative agent for vesicular disease in pigs is that the majority of the historical SVA isolates failed to reproduce the disease under experimental conditions [32,33]. In contrast, the contemporary SVA isolates can easily cause vesicular diseases in naturally or experimentally infected pigs [32], indicating that SVA is evolving towards a more virulent phenotype over time. As a newly emerging causative agent for pigs, exploring the pathogenic and immune escape mechanisms of SVA from diverse aspects will be conducive to finding potential antiviral targets or strategies. To date, a number of studies have demonstrated that there exist extremely complex interactions between SVA and autophagy/apoptosis in host cells [34,35]. For example, Sun and colleagues showed that SVA infection can induce autophagy in the early stage of SVA infection, which functions to inhibit SVA replication by degrading the SVA 3C protein; however, in the late stage of infection, SVA utilizes 2AB protein to inhibit autophagy via interaction with MARCHF8/MARCH8 and LC3 to facilitate viral replication [34]. Wen et al. (2021) found that, although selective autophagy receptor SQSTM1/p62 functions to inhibit SVA replication by targeting viral VP1 and VP3 to phagophores for an autophagic degradation, SVA has evolved an antagonistic mechanism against the function of selective autophagic degradation via cleavage of SQSTM1/p62 at glutamic acid 355, glutamine 392, and glutamine 395 by the SVA 3C protease (3C^pro^) [35]. Furthermore, it was demonstrated that both 2C and 3C^pro^ proteins of SVA are able to induce apoptosis, among which 2C protein induces apoptosis via the mitochondrion-mediated intrinsic pathway, while 3C^pro^ induces apoptosis through both the mitochondrial pathway and the extrinsic death receptor pathway [36]. Despite sustained efforts as mentioned above, the key factors that affect the replication, pathogenicity and virulence of SVA are far from being fully deciphered and thus warrant further investigation.

It is well known that quantitative proteomics is a powerful tool widely used for the systematic, large-scale and highly efficient analysis of the global protein profiles, protein function predictions, protein–protein interactions and signaling pathway analyses from a specific cell, tissue or organism [17,37]. In addition to TMT labeling, although other labeling approaches, such as an isobaric tag for relative and absolute quantitation (iTRAQ) and stable isotope labeling by amino acids in cell culture (SILAC), are also currently available for proteomic analysis, TMT is still the most commonly used method because it has several advantages over its counterparts, such as an increased sample multiplexing for relative quantitation, an enhanced sample throughput and fewer missing quantitative channels among samples. In the present study, in order to find more potential clues and targets for the investigation of SVA pathogenesis and immune escape mechanism, a quantitative proteomics approach based on TMT labeling coupled to LC-MS/MS was used for the comparative analysis of the dynamic changes of proteome in PK-15 cells in response to SVA infection. Although a variety of porcine-derived and non-porcine-derived cell lines have been demonstrated to be highly permissive to SVA infection, given that swine kidney is one of the natural target organs for SVA infection in vivo and SVA replicates well in swine kidneys [8,38], we, therefore, chose to use PK-15 for the quantitative proteomic analysis with the goal of obtaining experimental closer to the physiological state of pigs and the true state of SVA infection in vivo. Meanwhile, to depict a global picture of changes in host and viral proteins throughout the course of SVA infection, we specially set up three time points post-infection (12, 24 and 36 hpi) for sampling for a temporal proteomic analysis, by which a total of 8512 proteins were finally identified in both mock- and SVA-infected PK-15 cells, including 6979 quantified proteins and 1533 qualitative proteins (Appendix A). After a stringent filtering and quality check of the MS data, although six proteins (UniProt accession Nos. A0A2K9YSP5, A0A218L147, A0A5C2GZI4, A0A649YC94, A0A240FRZ3, A0A649YCR0, A0A649YCR7 and A0A6B9QIJ0) were annotated to belong to SVA, only two of them were identified to be VP0 and VP1 proteins due to the incomplete annotation for SVA proteins in the UniProt database (Appendix A). What needs to be pointed out is that VP0 is an important component of the immature capsid precursor protein. Upon proteolytic processing, VP0 will eventually be cleaved into two mature capsid proteins VP2 and VP4.

To ensure the reliability of the quantitative proteomic data, we randomly selected five proteins Mx1, eIF4E, G6PD, TOP1 and PGAM1 out of the screened DEPs for validation by qPCR and WB at the mRNA and protein level, respectively. In order to make the randomly selected DEPs more representative, one upregulated protein and one downregulated protein were, respectively, selected from each time point post-infection for the verification. Specifically, Mx1, Mx1 and TOP1 represent the upregulated proteins, while eIF4E, G6PD and PGAM1 represent the downregulated proteins at 12, 24 and 36 hpi, respectively. Mx1 is an IFN-induced dynamin-like GTPase that participates in cellular antiviral responses against a wide range of RNA viruses and some DNA viruses [27]. eIF4E is an important regulatory factor responsible for initiating protein synthesis via recognizing and binding to the 7-methylguanosine-containing mRNA cap [39]. G6PD is a cytosolic enzyme encoded by a housekeeping gene whose main physiologic role is to produce nicotinamide adenine dinucleotide phosphate (NADPH) [40]. TOP1 is a ubiquitous enzyme that modulates the topologic states of DNA during replication, transcription and chromosomal recombination [41]. PGAM1 is an important mutase that catalyzes the conversion of 3-phosphoglycerate to 2-phosphoglycerate in the glycolytic pathway [42]. On the basis of the qPCR and WB validation results for the five DEPs, we confirmed that our proteomics data are reliable enough for the subsequent multiple bioinformatics analyses.

To annotate the DEPs identified in PK-15 cells in response to SVA infection, GO, COG, KEGG and STRING analyses were successively performed on the DEPs. Although the functions of each bioinformatics analysis method we used are different, the results of which all reveal that the significantly enriched DEPs were mainly involved in a variety of host defense mechanisms, including the innate immune response (response to type I interferon), defense response to virus, negative regulation of viral life cycle, and antigen processing and presentation of endogenous antigen, etc. in the early and middle stages of SVA infection, but they turned to mainly participate in various metabolic processes, such as the carbohydrate metabolic process, carboxylic acid metabolic process, small molecule metabolic process, oxoacid metabolic process, hormone metabolic process, organic acid metabolic process, monocarboxylic acid metabolic process, in the late stage of SVA infection. By means of KEGG enrichment analysis, we further discovered that the significantly enriched DEPs at 12 and 24 hpi were mainly involved in the following innate immune response-related pathways: the NOD-like receptor signaling pathway, RIG-I-like receptor signaling pathway, complement and coagulation cascades, and the TNF signaling pathway. Our findings are consistent with a recent iTRAQ-based proteomics study which demonstrated that most DEPs were enriched in the innate immune response-related pathways, including the RIG-I-like receptor signaling pathway, NOD-like receptor signaling pathway and cytosolic DNA-sensing pathway, in SVA-infected PK-15 cells rather than IBRS-2 cells [10]. It should be noted that, although a relatively lower MOI of 0.5 than ours (MOI = 5) was used in this study, the cell samples used for the proteomic analysis were also collected at an early stage of SVA infection (6 hpi); however, this study did not analyze cell samples at other time points post-infection [10]. Similarly, Li and colleagues also showed that many innate immune-related proteins, such as Mx1, Mx2, ISG15, IFIT1, OAS1 and DDX58, were significantly upregulated in SVA-infected ST cells at 12 and 24 hpi by iTRAQ-coupled LC-MS/MS analysis, even though a different cell type and a lower MOI of 0.1 were used [43]. Furthermore, a transcriptome analysis also discovered that the innate immune-related genes and pathways were significantly activated in SVA-infected porcine renal proximal tubule epithelial cells (LLC-PK1) at 6 and 12 hpi, which were also in the early stage of SVA infection [44]. Interestingly, another proteomic analysis of SVA-infected kidney cells of golden hamsters (BSR-T7/5) using TMT-labeled nano-LC-MS/MS analysis found that the DEPs were mainly enriched in 11 cellular metabolism-related signaling pathways at 12 hpi, but no DEPs were enriched in the innate immune-related signaling pathways [45]. The possible reason for this inconsistency might be related to different species-derived cells being used for the proteomic analysis. Nonetheless, on the basis of these previous findings along with our experimental results, we conclude that, although the innate immune response can be successfully activated in the early stage after SVA invading the host cells, the virus has evolved multiple mechanisms to escape the host innate immune response in the late stage of viral infection. Thus far, a couple of studies have demonstrated that the 3C^pro^ protein of SVA is able to suppress type I IFN production via the cleavage of MAVS, TRIF, TANK or the degradation of IRF3 and IRF7, and that the protease activity of 3C^pro^ is required for its cleavage and degradation functions [13,46]. Xue et al. (2018) showed that the SVA 3C^pro^ protein can inhibit the ubiquitination of RIG-I, TBK1 and TRAF3 dependent on its deubiquitinating activity, thereby suppressing the type I interferon pathway [47].

Although it was shown that SVA has evolved diverse mechanisms to evade the innate immune response of the host in the late stage of infection, our proteomic data showed that a variety of ISG family proteins such as Mx1, Mx2, IFIT1, ISG15 and OASL were significantly upregulated in the early and middle stages of SVA infection. This indicates that the innate immunity still works during the course of SVA infection. To elucidate what role these ISG family proteins play during SVA infection of host cells, we assessed the effect of their gain- and loss-of-function on the replication of SVA in host cells, and discovered that all the five ISG family proteins can exert antiviral activities against SVA. As existing studies have shown that Mx1 and ISG15 usually exert antiviral activities by interacting with viral proteins [25,27,28], we took a step further by performing Co-IP and confocal microscopy on the co-transfected 293T cells, and found that Mx1 protein interacts with the VP1, VP2 and VP3 proteins of SVA, while ISG15 does not interact with any SVA proteins. Moreover, it was demonstrated that Mx1 protein possesses three important activities, including GTPase activity, oligomerization activity and interaction activity, each of which plays a crucial but not identical role in combating with different viruses [25,29,30]. Therefore, we constructed three Mx1 mutants GTPase-deficient Mx1 (K83A), oligomerization-disrupting Mx1 (R409D) and interaction-interfering Mx1 (ΔL4), and evaluated their ability to interact with VP1, VP2 and VP3 proteins. We finally proved that the three activities of Mx1 are indispensable for its interaction with VP1 and VP2 proteins, while the interaction of Mx1 with VP3 protein only depend on the oligomerization activity.

## Figures and Tables

**Figure 1 viruses-14-00863-f001:**
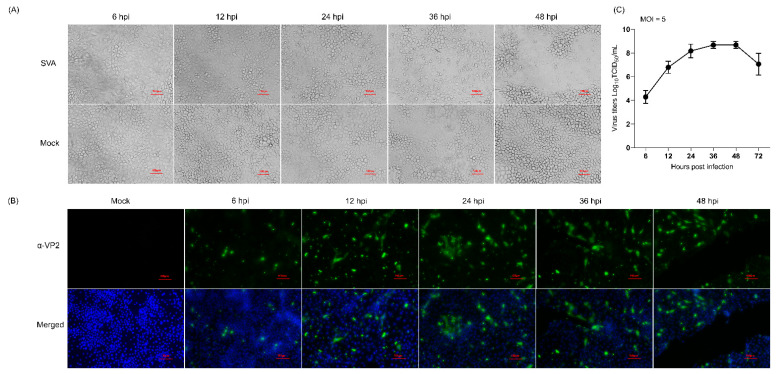
Characterization of the proliferation dynamics of Senecavirus A (SVA) in porcine kidney epithelial (PK-15) cells. (**A**) The cytopathic effect of PK-15 cells caused by SVA infection. PK-15 cells were mock-infected or infected with SVA SDta/2018 at an MOI of 5 50% tissue culture infectious dose (TCID_50_)/cell for 6, 12, 24 and 36 h, respectively. Scale bars, 10 μm. (**B**) Confirmation of SVA proliferation in PK-15 cells by indirect immunofluorescence assay (IFA) using the SVA VP2-specific monoclonal antibody (mAb) 2F5 (α-VP2) as the primary antibody. Cell nuclei were counterstained with 4′, 6-diamidino-2-phenylindole (DAPI). Scale bars, 100 μm. (**C**) One-step growth curve of SVA in PK-15 cells. The viral titer was determined by measuring the TCID_50_ using a microtitration assay. The data were recorded as means ± standard deviation (SD) from three independent experiments.

**Figure 2 viruses-14-00863-f002:**
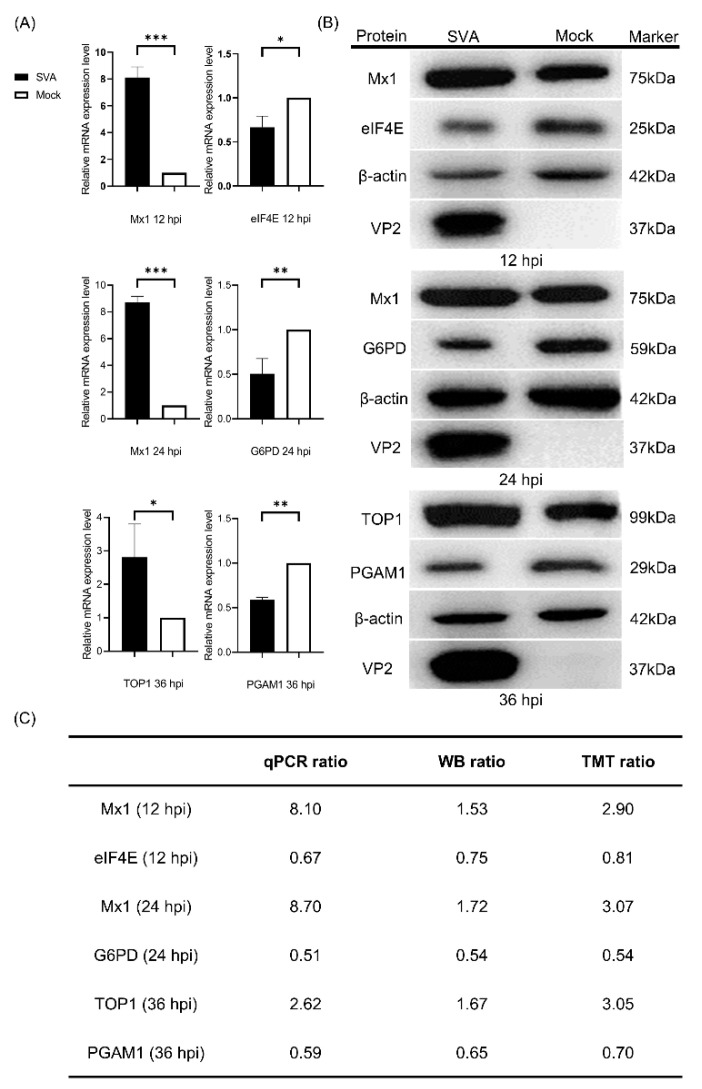
Validation of the differentially expressed proteins (DEPs) identified by tandem mass tags (TMT)-coupled LC-MS/MS analysis. (**A**) Quantitative real-time PCR (qPCR) analyses of the relative mRNA expression levels of Mx1, eukaryotic initiation factor 4E (eIF4E), glucose 6 phosphate dehydrogenase (G6PD), topoisomerase I (TOP1) and phosphoglycerate mutase 1 (PGAM1) in PK-15 cells in response to SVA infection. PK-15 cells were mock infected or infected with SVA SDta/2018 at an MOI of 5. At 12, 24 and 36 hpi, the cells were harvested and processed for qPCR analysis using β-actin gene as the internal control. Error bars indicate the standard error of three independent experiments (Student’s *t* test; * *p* < 0.05, ** *p* < 0.01 and *** *p* < 0.001). (**B**) Western blot (WB) analyses of the five representative DEPs in PK-15 cells in response to SVA infection. PK-15 cells were infected as in (**A**) and then subjected to WB analyses using rabbit anti-Mx1, -eIF4E, -G6PD, -TOP1 and -PGAM1 mAbs as the primary antibodies and with β-actin as the internal loading control. The SVA VP2-specific mAb 2F5 was used to track the progression of viral infection. The relative molecular weights of each target protein are indicated on their right side. (**C**) The relative mRNA ratio (qPCR ratio) and densitometric ratio (WB ratio) of the five representative DEPs between SVA- and mock-infected PK-15 cells was quantified and normalized to β-actin (WB ratio), which were compared with their respective ratios of the MS data (TMT ratio).

**Figure 3 viruses-14-00863-f003:**
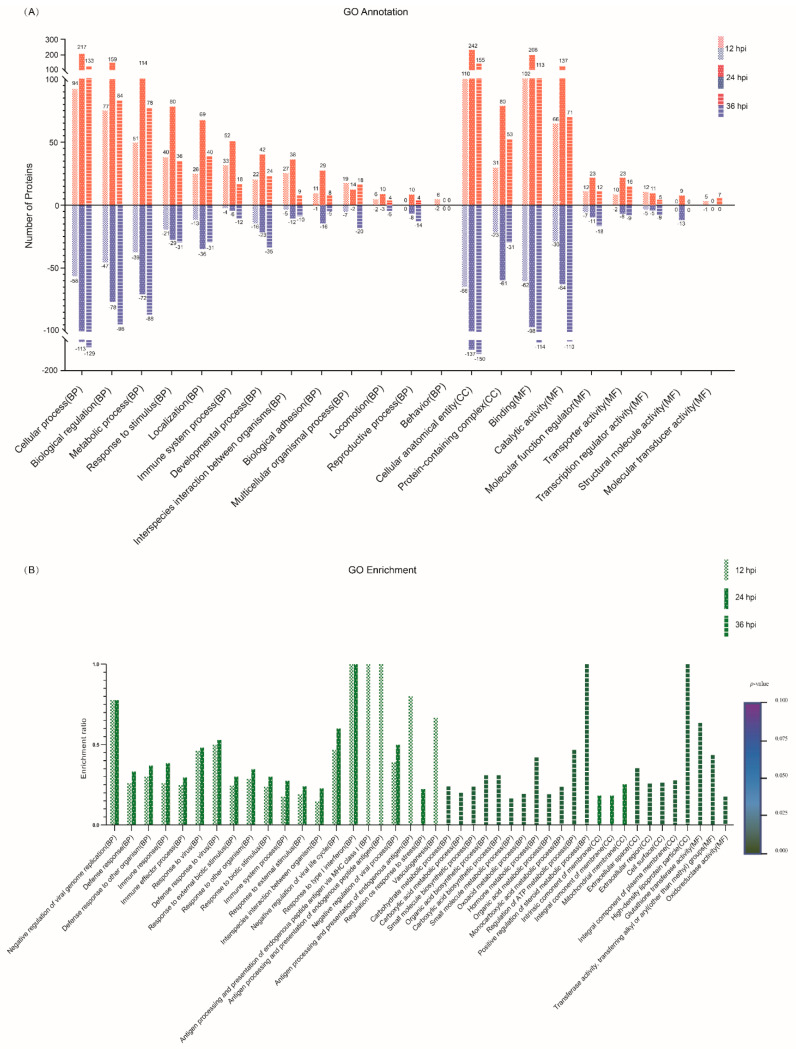
Gene ontology (GO) functional annotation for the DEPs identified in PK-15 cells in response to SVA infection. (**A**) The top 20 GO terms with the most DEPs at 12, 24 and 36 hpi, respectively. The relevant DEPs were classified into 22 subclasses under the three major GO categories of biological process (BP), cellular component (CC) and molecular function (MF). The abscissa texts indicate the name and classification of GO terms. The red and blue columns represent the upregulated and downregulated DEPs, respectively. The number of DEPs is marked on the top of each column. (**B**) The top 20 significantly enriched GO terms at 12, 24 and 36 hpi, respectively. The height of the column indicates the enrichment rate, and the color indicates the significance of the enrichment (*p* value). The darker the color, the more significant the enrichment of the GO term (Fisher’s exact test; *p* < 0.002).

**Figure 4 viruses-14-00863-f004:**
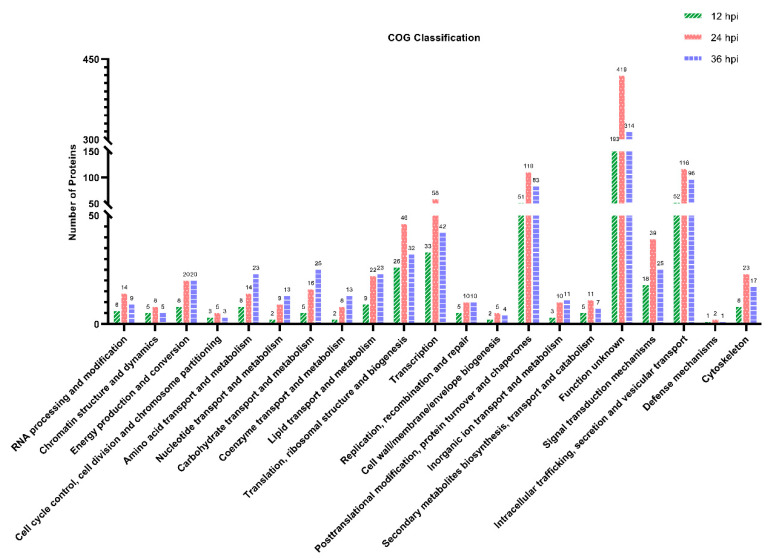
Clusters of Orthologous Groups (COG) function classification of the DEPs identified in PK-15 cells during SVA infection. A total of 445, 965 and 776 DEPs identified at 12, 24 and 36 hpi, respectively, were assigned to one or more of the 21 COG categories. The abscissa represents the COG categories and the ordinate indicates the number of DEPs in each category.

**Figure 5 viruses-14-00863-f005:**
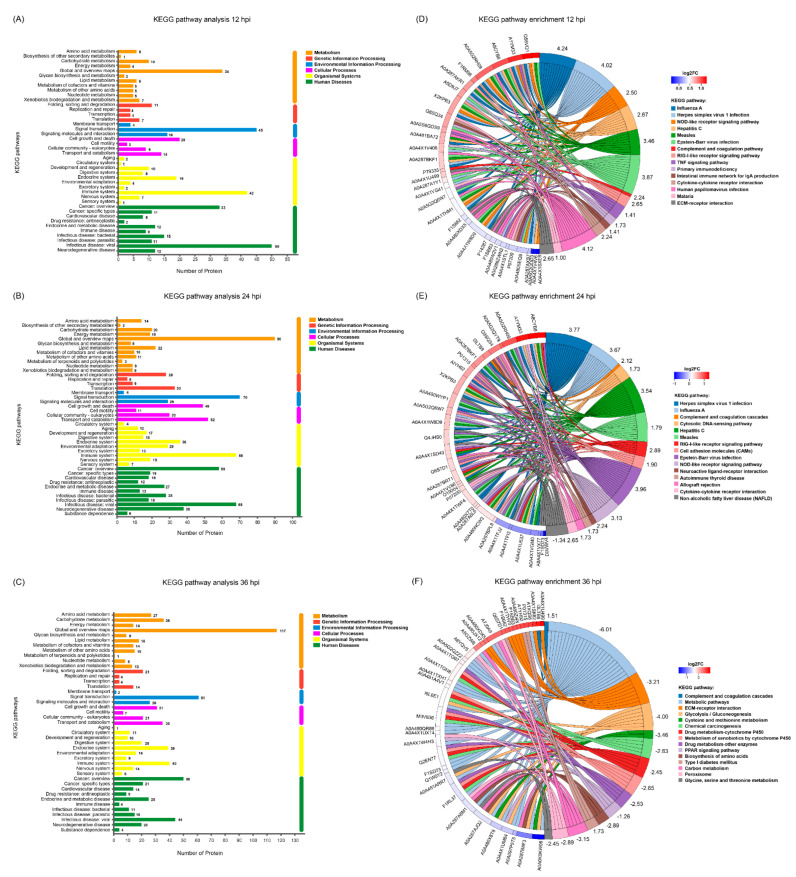
Kyoto Encyclopedia of Genes and Genomes (KEGG) pathway analysis of the DEPs identified in PK-15 cells during SVA infection. (**A**–**C**) KEGG pathway analyses for the DEPs identified at 12, 24 and 36 hpi, respectively. The different colors indicate 6 first-category KEGG pathways. The Y axis represents the name of each second-category KEGG pathway, and the X axis represents the number of proteins in each second-category. (**D**–**F**) KEGG pathway enrichment analyses for the DEPs identified at 12, 24 and 36 hpi, respectively. The UniProt accession numbers for the corresponding proteins are marked on the left of the chord diagrams. The log2 fold changes (log2FC) of the upregulated and downregulated proteins are marked with different colors between the red and blue gamut. The names of the top 15 KEGG signaling pathways are marked on the right of the chord diagram in 15 different colors. The Z-scores for the relevant KEGG signaling pathways are also marked on the right.

**Figure 6 viruses-14-00863-f006:**
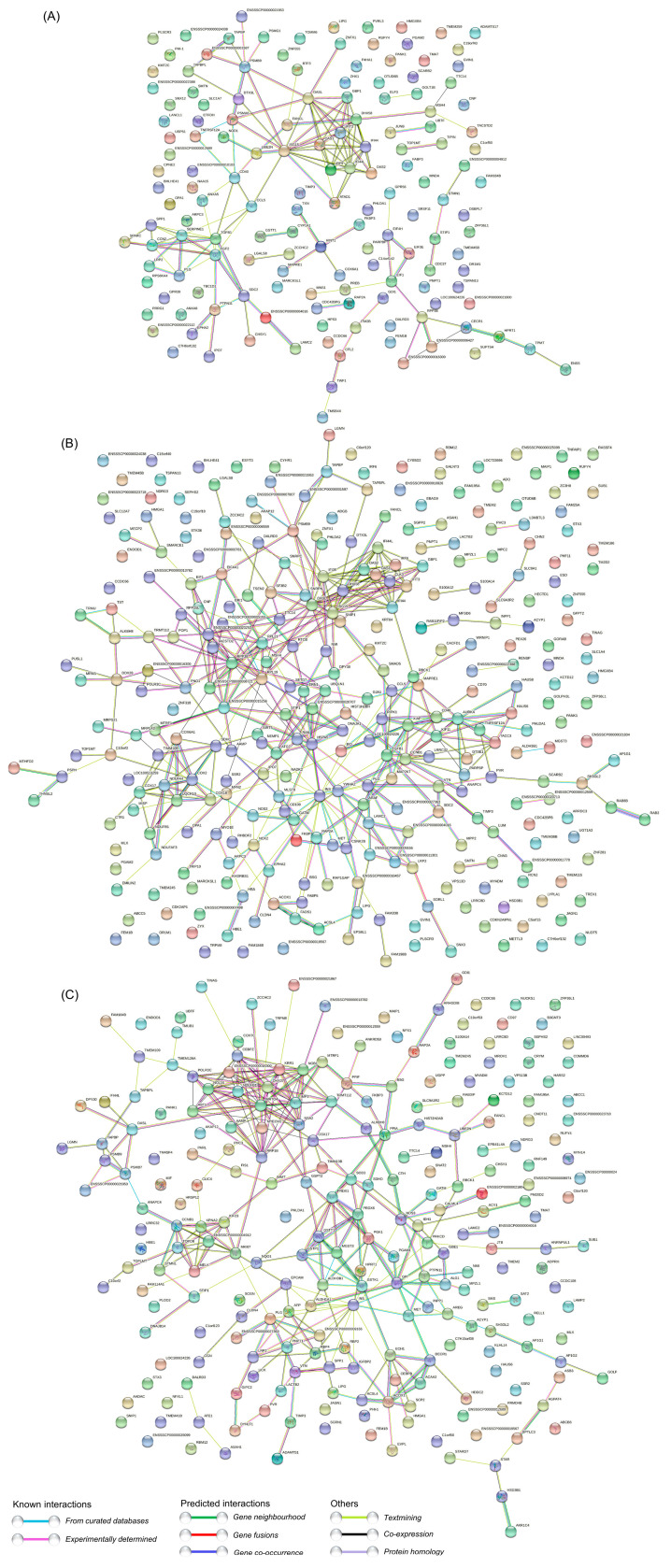
Protein–protein interaction (PPI) networks of the DEPs identified in PK-15 cells during SVA infection. (**A**) PPI networks of the DEPs at 12 hpi. (**B**) PPI networks of the DEPs at 24 hpi. (**C**) PPI networks of the DEPs at 36 hpi. The networks were constructed using the STRING database together with a medium confidence score of 0.4. The colored nodes represent query proteins and the first shell of interactors. The colored lines between two nodes represent known or predicted interactions and others. The edges represent protein–protein associations.

**Figure 7 viruses-14-00863-f007:**
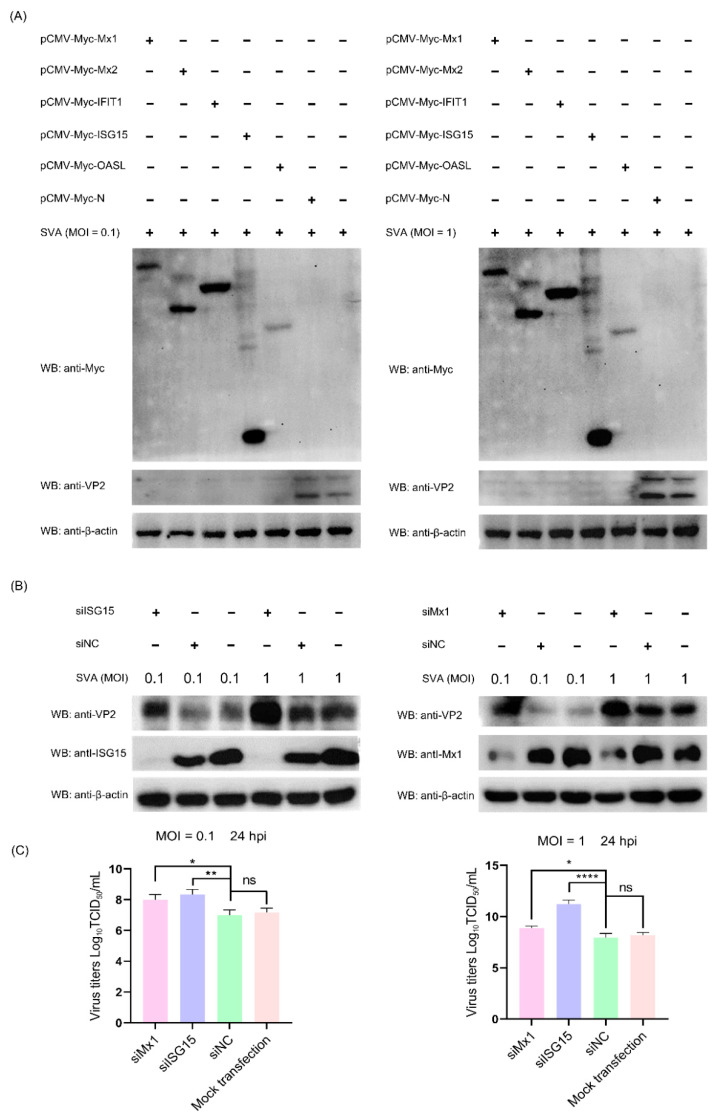
The interferon-stimulated gene (ISG) family proteins can efficiently inhibit the replication of SVA in PK-15 cells. (**A**) Overexpression of ISG proteins in PK-15 cells inhibits SVA replication. PK-15 cells were non-transfected or transfected with pCMV-Myc-Mx1, pCMV-Myc-Mx2, pCMV-Myc-IFIT1, pCMV-Myc-ISG15, pCMV-Myc-OASL or pCMV-Myc-N (empty vector). At 36 hpt, the transfected cells were infected with SVA at an MOI of 0.1 or 1. At 24 hpi, the cells were harvested and processed for WB analysis of VP2 protein expression. (**B**,**C**) Knockdown of ISG15 and Mx1 proteins in PK-15 cells promotes SVA replication. PK-15 cells were mock-transfected or transfected with ISG15-, Mx1-specifc siRNAs or the scrambled siRNAs (siNC). At 36 hpt, the transfected cells were infected with SVA at an MOI of 0.1 or 1. At 24 hpi, the cells and progeny viruses were harvested for WB analysis and viral yield titration, respectively. Data were expressed as means ± SD of three independent experiments (two-way ANOVA test; ns *p* > 0.05; * *p* < 0.05; ** *p* < 0.01; **** *p* < 0.0001).

**Figure 8 viruses-14-00863-f008:**
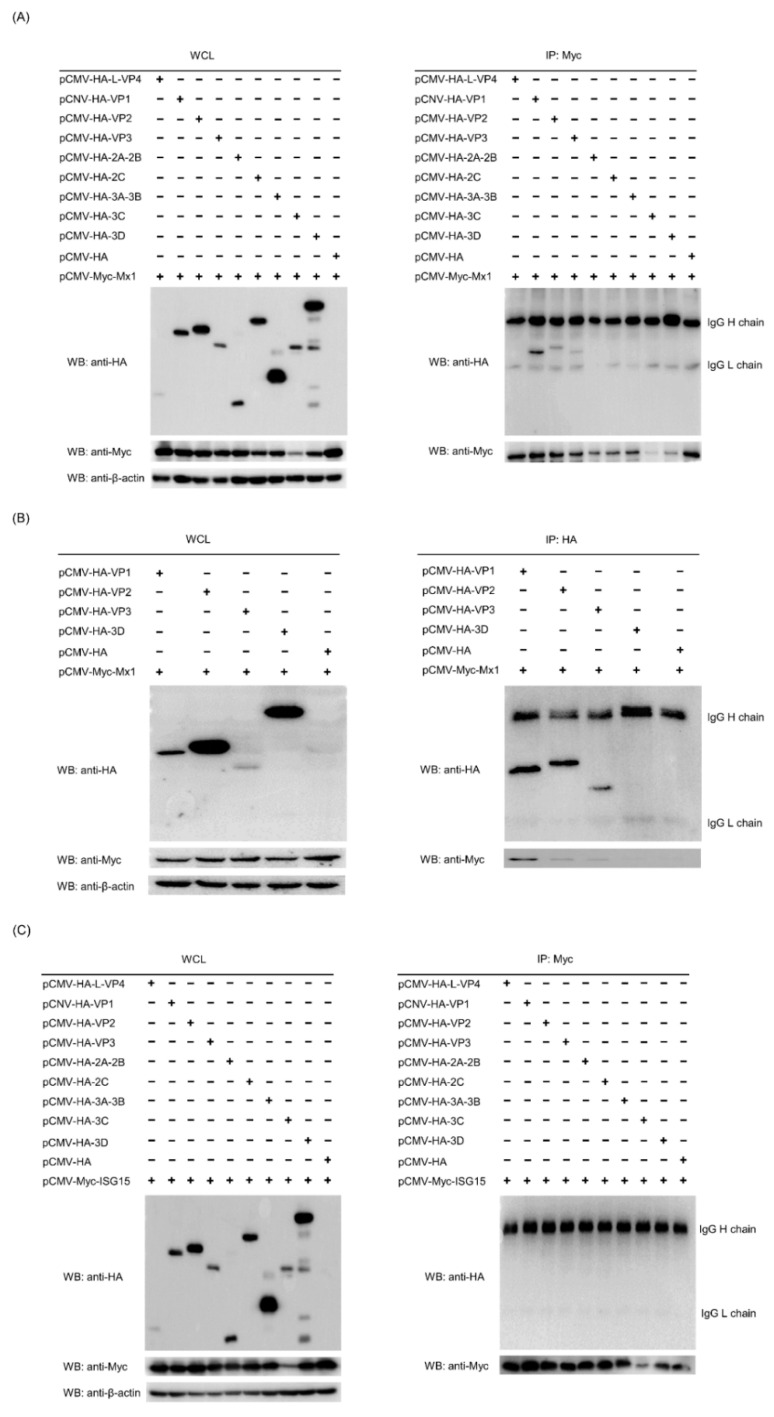
Mx1 but not ISG15 interacts with the VP1, VP2 and VP3 proteins of SVA. (**A**,**B**) WB and co-immunoprecipitation (Co-IP) analyses of HA-tagged SVA proteins and Myc-tagged Mx1 protein in co-transfected HEK293T cells. HEK293T cells were co-transfected with 1.25 μg of pCMV-HA-L-VP4, pCMV-HA-VP1, pCMV-HA-VP2, pCMV-HA-VP3, pCMV-HA-2A-2B, pCMV-HA-2C, pCMV-HA-3A-3B, pCMV-HA-3C, pCMV-HA-3D or pCMV-HA, and 1.25 μg of pCMV-Myc-Mx1. At 36 hpt, the cells were harvested to prepare whole cell lysates (WCL) for WB and Co-IP analyses using mouse anti-Myc and -HA mAbs. (**C**) WB and Co-IP analyses of HA-tagged SVA proteins and Myc-tagged ISG15 protein in co-transfected HEK293T cells. The cells were co-transfected, harvested, processed and analyzed as in (**A**), but with a different plasmid pCMV-Myc-ISG15.

**Figure 9 viruses-14-00863-f009:**
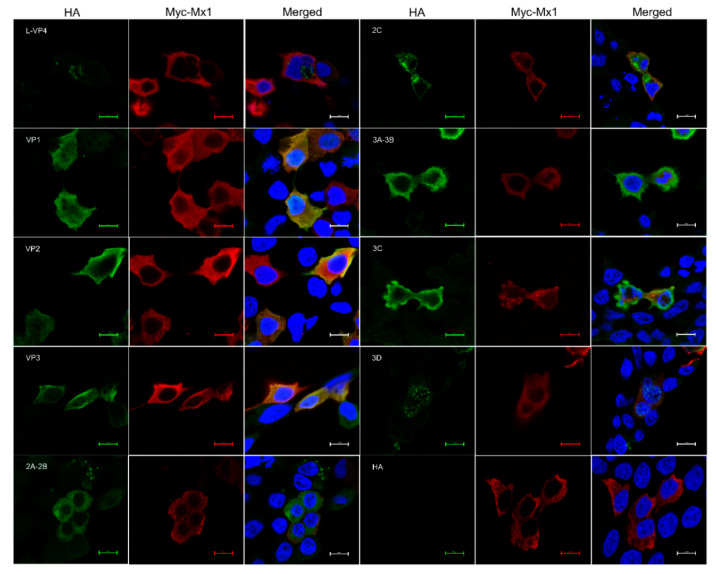
Confocal immunofluorescence microscopy of the colocalization of Mx1 and SVA proteins in co-transfected HEK293T cells. HEK293T cells were co-transfected with 1.25 μg of pCMV-HA-L-VP4, pCMV-HA-VP1, pCMV-HA-VP2, pCMV-HA-VP3, pCMV-HA-2A-2B, pCMV-HA-2C, pCMV-HA-3A-3B, pCMV-HA-3C, pCMV-HA-3D or pCMV-HA, and 1.25 μg of pCMV-Myc-Mx1. At 36 hpt, the cells were fixed for confocal microscopy using mouse anti-Myc and rabbit anti-HA mAbs as the primary antibodies. Scale bars, 10 μm.

**Figure 10 viruses-14-00863-f010:**
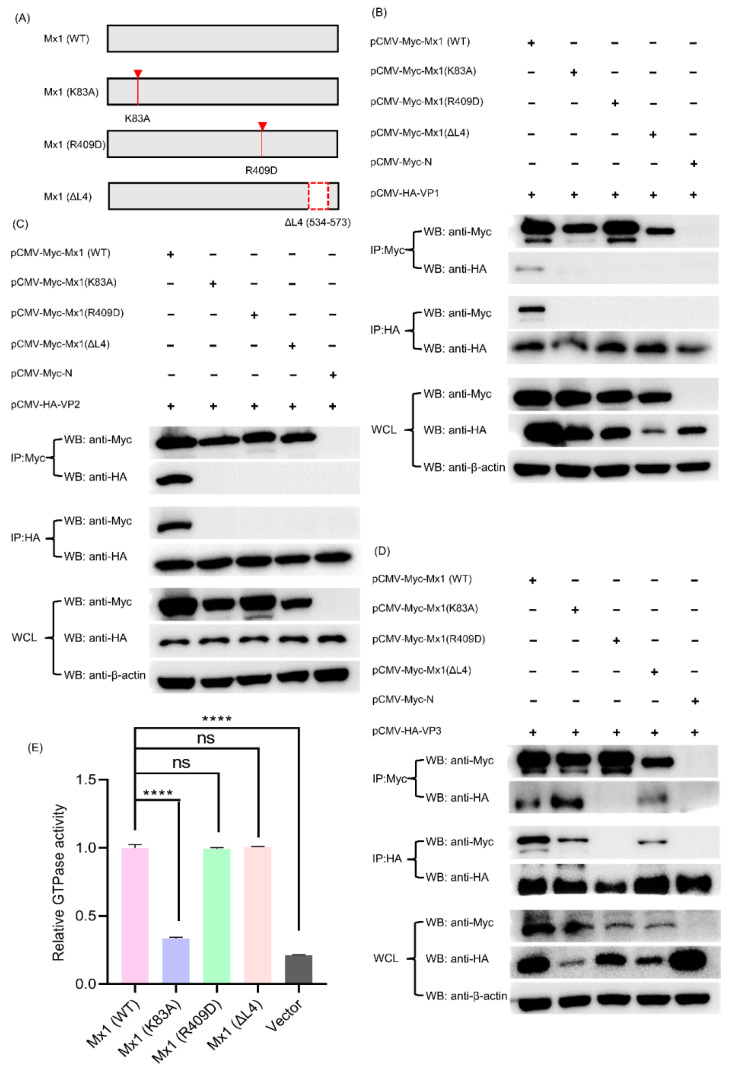
The GTPase, oligomerization and interaction activities of Mx1 are required for its interaction with the SVA VP1, VP2 and VP3 proteins. (**A**) Construction strategy of the three Mx1 mutants (Mx1 (K83A), Mx1 (R409D), and Mx1 (ΔL4)) with a single-site mutation of K83A, R409D, and a deletion of amino acid residues 534–573 (denoted by a red dashed box), respectively, using pCMV-Myc-Mx1 as the backbone plasmid. (**B**) Co-IP analysis of wild-type (WT) and mutant Mx1 proteins with SVA VP1 protein. (**C**) Co-IP analysis of WT and mutant Mx1 proteins with SVA VP2 protein. (**D**) Co-IP analysis of WT and mutant Mx1 proteins with SVA VP3 protein. HEK293T cells were co-transfected with 1.25 μg of pCMV-HA-VP1, pCMV-HA-VP2 or pCMV-HA-VP3, and 1.25 μg of pCMV-Myc-Mx1, pCMV-Myc-Mx1(K83A), pCMV-Myc-Mx1(R409D), pCMV-Myc-Mx1(ΔL4) or pCMV-Myc-N. At 36 hpt, the cells were harvested to prepare WCL for Co-IP analyses using mouse anti-Myc and -HA mAbs. (**E**) Determination of the GTPase activity of WT and mutant Mx1 proteins expressed in HEK293T cells. The cells were transfected with 2.5 μg of pCMV-Myc-Mx1, pCMV-Myc-Mx1(K83A), pCMV-Myc-Mx1(R409D), pCMV-Myc-Mx1(ΔL4), pCMV-Myc-ISG15, or pCMV-Myc-N (empty vector). At 36 hpt, the cells were harvested, lysed and processed for GTPase activity assay using a QuantiChrom^™^ ATPase/GTPase assay kit. Data were expressed as means ± SD of three independent assays (two-way ANOVA test; ns *p* > 0.05; **** *p* < 0.0001).

**Table 1 viruses-14-00863-t001:** The primers used in this study.

Plasmids Name	Primers	Sequence (5′-3′)	Restriction Enzymes	GenBank Accession Numbers
pCMV-Myc-Mx1	pCMV-Myc-Mx1-F	GCCATGGAGGCCCGAATTCGGATGGTTTATTCCAACTGTGAAAG	EcoR I	DQ095779.1
	pCMV-Myc-Mx1-R	GCGGCCGCGGTACCTCGAGTCAGCCTGGGAACTTGG	Xho I	
pCMV-Myc-Mx2	pCMV-Myc-Mx2-F	GCCATGGAGGCCCGAATTCGGATGCCTAAACCCCGCAT	EcoR I	NM_001097416.1
	pCMV-Myc-Mx2-R	GCGGCCGCGGTACCTCGAGTTACATCCCTTGTACCTCAACC	Xho I	
pCMV-Myc-ISG15	pCMV-Myc-ISG15-F	GCCATGGAGGCCCGAATTCGGATGGGTAGGGAACTGAAGGT	EcoR I	EU647216.1
	pCMV-Myc-ISG15-R	GCGGCCGCGGTACCTCGAGCTAGCACTCGGTGGGGT	Xho I	
pCMV-Myc-OASL	pCMV-Myc-OASL-F	GCCATGGAGGCCCGAATTCGGATGGAGCTATTTTACACCCCAG	EcoR I	MG679809.1
	pCMV-Myc-OASL-R	GCGGCCGCGGTACCTCGAGTCAGTCACAGCCTTTGGCT	Xho I	
pCMV-Myc-IFIT1	pCMV-Myc-IFIT1-F	GCCATGGAGGCCCGAATTCGGATGAGTAATAATGCTGATGAAGATCAG	EcoR I	JN621781.1
	pCMV-Myc-IFIT1-R	GCGGCCGCGGTACCTCGAGTTAGGGATCAAGTCCCTCAGATT	Xho I	
pCMV-HA-L-VP4	pCMV-HA-L-VP4-F	GCCATGGAGGCCCGAATTCGGATGCAGAACTCTCATTTTTCTTTC	EcoR I	MN433300.1
	pCMV-HA-L-VP4-R	GCGGCCGCGGTACCTCGAGTCATTTGAGGTAGCCAAGAGGGTT	Xho I	
pCMV-HA-VP1	pCMV-HA-VP1-F	GCCATGGAGGCCCGAATTCGGATGTCCACCGACAACGCC	EcoR I	MN433300.1
	pCMV-HA-VP1-R	GCGGCCGCGGTACCTCGAGTCATTGCATCAGCATCTTCTGC	Xho I	
pCMV-HA-VP2	pCMV-HA-VP2-F	GCCATGGAGGCCCGAATTCGGATGGATCACAATACCGAAGAAATGG	EcoR I	MN433300.1
	pCMV-HA-VP2-R	GCGGCCGCGGTACCTCGAGTCACTGTTCCTCGTCCGTCC	Xho I	
pCMV-HA-VP3	pCMV-HA-VP3-F	GCCATGGAGGCCCGAATTCGGATGGGGCCCATTCCCAC	EcoR I	MN433300.1
	pCMV-HA-VP3-R	GCGGCCGCGGTACCTCGAGTCAGTGGAACACGTAGGAAGGATT	Xho I	
pCMV-HA-2A-2B	pCMV-HA-2A-2B-F	GCCATGGAGGCCCGAATTCGGATGTCAGGCGACATCGAGAC	EcoR I	MN433300.1
	pCMV-HA-2A-2B-R	GCGGCCGCGGTACCTCGAGTCATTGCATCTTGAACAGCTTTC	Xho I	
pCMV-HA-2C	pCMV-HA-2C-F	GCCATGGAGGCCCGAATTCGGATGGGACCCATGGACACAGTC	EcoR I	MN433300.1
	pCMV-HA-2C-R	GCGGCCGCGGTACCTCGAGTCACTGTAGAACCAGAGTCTGCATATTTC	Xho I	
pCMV-HA-3A-3B	pCMV-HA-3A-3B-F	GCCATGGAGGCCCGAATTCGGATGAGCCCTAACGAGAACGACG	EcoR I	MN433300.1
	pCMV-HA-3A-3B-R	GCGGCCGCGGTACCTCGAGTCATTGCATTTCCATAAGAGAG	Xho I	
pCMV-HA-3C	pCMV-HA-3C-F	GCCATGGAGGCCCGAATTCGGATGCAGCCCAACGTGGACAT	EcoR I	MN433300.1
	pCMV-HA-3C-R	GCGGCCGCGGTACCTCGAGTCATTGCATTGTAGCCAGAGGC	Xho I	
pCMV-HA-3D	pCMV-HA-3D-F	GCCATGGAGGCCCGAATTCGGATGGGACTGATGACTGAGCTAGAGC	EcoR I	MN433300.1
	pCMV-HA-3D-R	GCGGCCGCGGTACCTCGAGTCAGTCGAACAAGGCCCT	Xho I	
pCMV-Myc-Mx1(K83A)	pCMV-Myc-Mx1(K83A)-F	CAGTTCGGGCGCGAGCTCCGTGCTGGAGGCCCT		DQ095779.1
	pCMV-Myc-Mx1(K83A)-R	AGCACGGAGCTCGCGCCCGAACTCTGGTCCCCGAT		
pCMV-Myc-Mx1(R409D)	pCMV-Myc-Mx1(R409D)-F	TACCAAGATGGATAATGAGTTCTGCAAATGGAGTGC		DQ095779.1
	pCMV-Myc-Mx1(R409D)-R	AGAACTCATTATCCATCTTGGTAAACAGCCGACACT		
pCMV-Myc-Mx1(ΔL4)	pCMV-Myc-Mx1(ΔL4)-F	ATCGTGTACTCCATAGCCGAGATCTTTCAGCAC		DQ095779.1
	pCMV-Myc-Mx1(ΔL4)-R	GGCTATGGAGTACACGATCTGCTCCATTTGGAAC		
	qMx1-F	GGCGTGGGAATCAGTCATG		NM_214061.2
	qMx1-R	AGGAAGGTCTATGAGGGTCAGATCT		
	qTOP1-F	GGCCACCAGTGGAAGGA		XM_013993526.2
	qTOP1-R	CTCGTCCACCACGCCTT		
	qeIF4E-F	GCCTGGCTGTGACTACTCAC		XM_005656555.3
	qeIF4E-R	TCCAATAAGGCACAGCAGTG		
	qG6PD-F	CGCAACTCCTACGTGGC		XM_003360515.5
	qG6PD-R	GCGGATGTTCTTGGTGAC		
	qPGAM1-F	TACAAGCTGGTGCTGATCC		XM_003483535.4
	qPGAM1-R	CTGCACTGAGGTGAAGCAG		
	qβ-actin-F	TCCCTGGAGAAGAGCTACGA		XM_003124280.5
	qβ-actin-R	AGCACCGTGTTGGCGTAGAG		

**Table 2 viruses-14-00863-t002:** The specific and scrambled siRNAs used for the knockdown of Mx1 and ISG15 proteins in PK-15 cells.

siRNAs Name	Sense (5′-3′)	Antisense (5′-3′)	GenBank Accession Numbers	Sources
siMx1-216	CCAAUCACCUGUUACUAAATT	UUUAGUAACAGGUGAUUGGTT	DQ095779.1	Designed in this study
siMx1-732	GCAAUCAGCCAUACGACAUTT	AUGUCGUAUGGCUGAUUGCTT		Designed in this study
siMx1-1906	GCACCUGAUUGCCUACCAUTT	AUGGUAGGCAAUCAGGUGCTT		Designed in this study
siISG15	CUAUGUGCACCGUGUAUAUTT	AUAUACACGGUGCACAUAGTT	EU647216.1	[25]
siNC	UUCUCCGAACGUGUCACGUTT	ACGUGACACGUUCGGAGAATT	N/A	Designed in this study

## Data Availability

All the datasets supporting the conclusions of this article are included within the article or Appendix A. The obtained mass spectrometry proteomics data have been deposited to the ProteomeXchange Consortium (http://www.proteomexchange.org, accessed on 26 January 2022) via the Proteomics Identifications (PRIDE) partner repository with the dataset identifier PXD031260.

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
