# Peer review of "Comparative Proteomic Analysis Reveals Mx1 Inhibits Senecavirus A Replication in PK-15 Cells by Interacting with the Capsid Proteins VP1, VP2 and VP3"

_viruses, 2022, doi:10.3390/v14050863_

Round 1

Reviewer 1 Report

The study titled 'Comparative Proteomic Analysis Reveals Mx1 Inhibits Sene- 2 cavirus A Replication in PK-15 Cells by Interacting with the 3 Capsid Proteins VP1, VP2 and VP3' by Gao et al., is overall a well controlled study. There are however some specific concerns:

  1. The authors should make sure they mention the figure number and panel when describing the results.
  2. In figure 7C in the virus titration graphs mock also chows signals. Can the authors explain that? 
  3. What is the difference between the left and right panel of figure 7A and B?
  4. For the benefit of the readers can the authors indicate all the mutations of Mx1 in a schematic representation.

Reviewer 2 Report

The manuscript entitled “Comparative proteomic analysis reveals Mx1 inhibits Senecavirus A replication in PK-15 cells by interacting with the capsid proteins VP1, VP2 and VP3” (manuscript ID: viruses-1676037) describes the quantitative temporal proteomic analysis of Senecavirus A (SVA)-infected porcine kidney epithelial PK-15 cells at the early, middle and late time points post-infection using the tandem mass tags (TMT) labeling coupled with liquid chromatography-tandem mass spectrometry (LC-MS/MS) method. After validating the proteomic data by quantitative PCR and western blot analyses of five randomly selected differentially expressed proteins (DEPs), all the DEPs were analyzed by diverse bioinformatics analyses, such as GO, COG, KEGG and STRING. Overall, the topic of the study is interesting and the proteomics methods used are adequate. The resulting proteomic data provide useful clues for further exploration of SVA. My specific comments are listed below:

  1. At present, several proteomic techniques are currently available for the reliable and high-quality quantification of proteome, such as tandem mass tags (TMT) labeling, isobaric tag for relative and absolute quantitation (iTRAQ) labeling, and stable isotope labeling by amino acids in cell culture (SILAC). Therefore, what is the reason why the authors chose to use TMT labeling coupled to LC-MS/MS technique? The authors at least discuss this in the Discussion section.
  2. L89-91: The authors used two different kinds of monoclonal antibodies (mAbs) against the hemagglutinin (HA) tag in their study. One is the rabbit anti-HA mAb purchased from Sigma-Aldrich, the other is mouse anti-HA mAb purchased from Medical & Biological Laboratories Co., Ltd. Please explain why two different mAbs were simultaneously used in the same study.
  3. The siRNA duplexes used for the specific knockdown of ISG15 were directly synthesized according to one of authors’ previous studies. I suggest listing the sources of the siMx1-, siISG- and siNC-siRNA duplexes directly in Table 2.

Reviewer 3 Report

This manuscript obtained abundant information about proteome dynamic changes in PK-15 cells during SVA infection at 12, 24 and 36 hpi. The research revealed that the DEPs were mainly involved in host innate and adaptive immune responses or participated in various metabolic processes, which provides valuable clues for further investigation of SVA pathogenesis.

This research was designed and performed logically, and the manuscript was written well. While, the following issues should be addressed before publishment.

1. The representation of some numbers in the manuscript should be clarified. For example, the number of 12000 on line 132 in page 3 should be represented as 12,000.

2. On line 81 in page 2, the proper noun CO2 should be represented as CO2.

3. On line 282, the phrase of small interfering RNAs (s) should be abbreviated as siRNA within brackets.

4. On line 622, the hpt was worse and should be corrected.

5. In the Figure 7 (c), the marker “ns” seems to have been mislabeled, please recheck it.

6. The reference of 19 seems too early, it can be considered to be changed with a paper not too long ago.

Reviewer 4 Report

In the manuscript (#viruses-1676037), Gao et al. performed a quantitative proteomic analysis of Senecavirus A (SVA)-infected PK-15 cells using TMT labeling coupled to LC−MS/MS technique. The analyses were done on an early, middle and late time points post SVA infection and were followed by multiple bioinformatics analyses of the identified differentially expressed cellular proteins (DEPs) using GO, COG, KEGG and STRING annotation tools. Finally, the authors investigated the antiviral activity of Mx1 protein against SVA. The novelty of the study lies in the virus being studied and a temporal proteomic analysis was conducted. The results were presented appropriately and easy to understand, although there are some minor concerns should be addressed. My specific comments on the manuscript are listed below:

  1. L753-754: VP0 and VP1 proteins were identified to belong to SVA; however, the authors described in the Introduction section (L34-36): “the polyprotein precursor is subsequently cleaved into 12 mature proteins, including the structural proteins VP1, VP2, VP3 and VP4. Obviously, the viral proteins of SVA do not contain VP0 protein, please explain this issue.
  2. As the authors mentioned in the Discussion section, several proteomic analyses were already performed on SVA-infected cells, however, the authors didn’t mention what kind of proteomic techniques were used the relevant studies.
  3. L477-478: Figure 4: The difference between the three columns’ colors to display different time points is imperceptible. Please update Figure 4 using distinctly different colors.
